# Epithelial magnesium transport by TRPM6 is essential for prenatal development and adult survival

Vladimir Chubanov[1]*, Silvia Ferioli[1], Annika Wisnowsky[1], David G Simmons[2], Christin Leitzinger[3], Claudia Einer[3], Wenke Jonas[4,5], Yuriy Shymkiv[6], Harald Bartsch[7], Attila Braun[8,9], Banu Akdogan[1], Lorenz Mittermeier[1], Ludmila Sytik[1], Friedrich Torben[10], Vindi Jurinovic[11], Emiel PC van der Vorst[12], Christian Weber[12,13], Önder A Yildirim[14,15], Karl Sotlar[7†], Annette Schürmann[4,5], Susanna Zierler[1], Hans Zischka[3], Alexey G Ryazanov[6,16], Thomas Gudermann[1,13,17]*

[1]Walther-Straub Institute of Pharmacology and Toxicology, Ludwig Maximilian University of Munich, Munich, Germany; [2]School of Biomedical Sciences, The University of Queensland, Brisbane, Australia; [3]Institute of Molecular Toxicology and Pharmacology, Helmholtz Zentrum Munich, Neuherberg, Germany; [4]Department of Experimental Diabetology, German Institute of Human Nutrition, Potsdam-Rehbruecke, Germany; [5]German Center for Diabetes Research, Munich, Germany; [6]Princeton Institute of Life Sciences, Princeton, United States; [7]Institute of Pathology, Ludwig Maximilian University of Munich, Munich, Germany; [8]Rudolf Virchow Center for Experimental Biomedicine, University of Würzburg, Würzburg, Germany; [9]Department of Vascular Medicine, University Hospital Würzburg, Würzburg, Germany; [10]Genome Analysis Center, Institute of Experimental Genetics, Helmholtz Zentrum Munich, Neuherberg, Germany; [11]Institute for Medical Informatics, Biometry and Epidemiology, Ludwig Maximilian University of Munich, Munich, Germany; [12]Institute for Cardiovascular Prevention, Ludwig Maximilian University of Munich, Munich, Germany; [13]German Centre for Cardiovascular Research, Munich Heart Alliance, Munich, Germany; [14]Comprehensive Pneumology Center, Institute of Lung Biology and Disease, Helmholtz Zentrum Munich, Neuherberg, Germany; [15]German Center for Lung Research, Munich, Germany; [16]Department of Cellular and Molecular Pharmacology, Rutgers Robert Wood Johnson Medical School, Piscataway, United States; [17]Comprehensive Pneumology Center Munich, German Center for Lung Research, Munich, Germany

*For correspondence: vladimir. chubanov@lrz.uni-muenchen.de (VC); thomas.gudermann@lrz.uni-muenchen.de (TG)

Present address: †Institute of Pathology, Paracelsus Medical University, Salzburg, Austria

Competing interests: The authors declare that no competing interests exist.

**Abstract** $Mg^{2+}$ regulates many physiological processes and signalling pathways. However, little is known about the mechanisms underlying the organismal balance of $Mg^{2+}$. Capitalizing on a set of newly generated mouse models, we provide an integrated mechanistic model of the regulation of organismal $Mg^{2+}$ balance during prenatal development and in adult mice by the ion channel TRPM6. We show that TRPM6 activity in the placenta and yolk sac is essential for embryonic development. In adult mice, TRPM6 is required in the intestine to maintain organismal $Mg^{2+}$ balance, but is dispensable in the kidney. *Trpm6* inactivation in adult mice leads to a shortened lifespan, growth deficit and metabolic alterations indicative of impaired energy balance. Dietary $Mg^{2+}$ supplementation not only rescues all phenotypes displayed by *Trpm6*-deficient adult mice,

but also may extend the lifespan of wildtype mice. Hence, maintenance of organismal $Mg^{2+}$ balance by TRPM6 is crucial for prenatal development and survival to adulthood.

## Introduction

$Mg^{2+}$ is the most abundant intracellular divalent cation and is essential for the regulation of a broad spectrum of metabolic and signalling pathways (*de Baaij et al., 2015*). In addition, direct association with $Mg^{2+}$ fosters the structural integrity of key metabolites (such as ATP), proteins, lipid membranes and nucleic acids (*de Baaij et al., 2015*) implying that organismal $Mg^{2+}$ deficiency, a surprisingly common condition in humans (*King et al., 2005*; *Rosanoff et al., 2012*), may be especially harmful during prenatal development and early postnatal life, when the production of and the demand for $Mg^{2+}$-bound metabolites is particularly high. There is growing evidence to suggest that $Mg^{2+}$ deprivation is accompanied by different types of metabolic, immune, cardiovascular and neurological disorders (*de Baaij et al., 2015*). However, mainly due to the lack of adequate mammalian genetic models, it still remains unclear whether an imbalance in $Mg^{2+}$ metabolism is merely associated with or can directly trigger the latter pathophysiological processes. Furthermore, it has recently been shown that cellular $Mg^{2+}$ fluxes regulate the circadian rhythm and energy balance (*Feeney et al., 2016*), CGRP-mediated osteogenic differentiation (*Zhang et al., 2016*) and synaptic plasticity (*Palacios-Prado et al., 2014*), and that changes in the composition of brain interstitial $Mg^{2+}$-concentrations participate in the control of the sleep-wake cycle (*Ding et al., 2016*).

The remarkable recent progress in our understanding of the critical role of $Mg^{2+}$ in health and disease contrasts with the dearth of knowledge about the mechanisms governing cellular and organismal $Mg^{2+}$ balance. Approximately 10 plasma membrane $Mg^{2+}$ channels have been proposed (*Quamme, 2010*) indicating a high degree of redundancy. However, quite some controversy surrounds the biological role of many of these proteins, and the question whether there is a central gatekeeper responsible for organismal $Mg^{2+}$ balance has not yet been answered. The kinase-coupled ion channel TRPM7 has been proposed as a ubiquitous, indispensable cellular $Mg^{2+}$ entry pathway (*Schmitz et al., 2003*; *Chubanov et al., 2004*; *Ryazanova et al., 2010*; *Stritt et al., 2016*). However, studies with *Trpm7* gene-deficient mice failed to confirm a corresponding in vivo role of *Trpm7*. Thus, constitutive inactivation of *Trpm7* in mice entailed early embryonic lethality for as yet unknown reasons (*Jin et al., 2008*). Furthermore, tissue-specific deletions of *Trpm7* in mouse embryos affected morphogenesis of internal organs apparently in a $Mg^{2+}$-independent manner (*Jin et al., 2008*, *2012*; *Sah et al., 2013*). More recently, it was suggested that the $Mg^{2+}$ transporter MagT1 rather than TRPM7 might play a critical role for $Mg^{2+}$ homeostasis in T lymphocytes (*Li et al., 2011*) and probably also in the whole embryo (*Zhou and Clapham, 2009*). Hence, the biological role of TRPM7 requires further clarification.

In the present work, we focussed on the closest TRPM7 relative, TRPM6, because loss-of-function mutations in *TRPM6* cause hypomagnesemia (low $Mg^{2+}$ blood levels) in human infants thought to mainly result from renal $Mg^{2+}$ wasting (*Schlingmann et al., 2002*; *Walder et al., 2002*; *Voets et al., 2004*). However, deletion of *Trpm6* in mice has resulted in neural tube closure defects and embryonic death (*Walder et al., 2009*) indicating a direct role of TRPM6 in developmental processes and calling into question the simplistic view on the human *TRPM6* phenotype.

By integrating systematic phenotyping of *Trpm6* gene-modified mice with biochemical analysis, gene expression, metabolomics, and cell biological approaches, we decipher the molecular and organismal roles of TRPM6 in prenatal development and postnatal survival.

## Results

### TRPM6 function in extraembryonic cells is essential for fetal development

To understand the role of *Trpm6* in prenatal development, we determined the onset of embryonic lethality in *Trpm6* null embryos and investigated the expression pattern of *Trpm6* at this stage. Using a mouse strain carrying a gene-trap mutation in *Trpm6* (*Trpm6$^{\beta geo}$*) (*Table 1*), we found that *Trpm6$^{\beta geo/\beta geo}$* embryos were present at embryonic days (e) 8.5–10.5 (*Figure 1A*). However, only

**eLife digest** A balanced diet contains a variety of minerals such as magnesium ions, which are required for many chemical reactions in our body. A shortage of magnesium ions is linked to many diseases and is thought to be especially harmful to babies in the womb and shortly after birth. Magnesium ion deficiency is widespread in human populations and in the US is thought to affect up to 68% of people.

Despite its prominent role in human health, our understanding of how the body maintains the right balance of magnesium ions remains extremely vague. Magnesium ions can enter and leave a cell by passing through specific types of proteins that form channels in the membrane surrounding the cell. There are thought to be around ten types of these magnesium ion channels in human cells, but we do not know what roles any of them perform in the body. One such channel called TRPM6 may be particularly important because mutations in the gene that encodes this channel can cause magnesium ion deficiency in human infants. However, the loss of TRPM6 in mice disrupts how mouse embryos develop, suggesting that our current view on the role that TRPM6 plays in regulating the magnesium ion balance in humans may be too simplistic.

To address this question, Chubanov et al. studied mice with mutations that disrupted the production of TRPM6 in specific tissues only. The experiments show that TRPM6 primarily operates in the placenta and intestine to regulate the balance of magnesium ions in the body. Further experiments show that the loss of TRPM6 in adult mice leads to reduced lifespan, growth defects and poor health by disrupting important biochemical reactions. Supplying the mutant mice with magnesium ion supplements improved their health and could extend lifespans of normal animals.

The findings of Chubanov et al. demonstrate that TRPM6 plays a crucial role in regulating the levels of magnesium ions in mice before birth and into adulthood. The next step is to carry out large-scale experiments to investigate the effects of altering the levels of magnesium ions in human diets.

a few mutants were found between e11.5–12.5 and no $Trpm6^{\beta geo/\beta geo}$ individuals were viable after e14.5 (*Figure 1A*). Compared to e9.5 C-shaped $Trpm6^{+/+}$ individuals, all $Trpm6^{\beta geo/\beta geo}$ embryos isolated had not turned (S-shaped) and were smaller indicating a developmental retardation after e8.5 (*Figure 1B*). Consequently, we investigated the expression pattern of *Trpm6* in e8.5 fetuses by in situ hybridization (ISH) and found that *Trpm6* was specifically expressed in the visceral yolk sac endoderm and extraembryonic chorion (*Figure 1C*) and that *Trpm6* was not detectable in the neural tube (*Figure 1—figure supplement 1*). Within the placental labyrinth a network of maternal sinus-oids are intertwined with fetal blood capillaries, separated by two layers of transporting trophoblast cells, syncytiotrophoblasts I (SynT-I) and II (SynT-II) (*Simmons and Cross, 2005*; *Simmons et al., 2008*). At e8.5, morphogenesis of the labyrinth is in the initial stages and SynT-I/SynT-II cell layers are distinguishable (*Simmons and Cross, 2005*; *Simmons et al., 2008*). We observed that *Trpm6* expression was restricted to SynT-I cells (*Figure 1D*). In the fully maturated labyrinth at e14.5 *Trpm6* mRNA was detected in syncytiotrophoblasts as well (*Figure 1E*).

Syncytiotrophoblasts and endoderm cells of the yolk sac exchange metabolites between the maternal and fetal blood (*Simmons and Cross, 2005*). To clarify whether TRPM6 is required for $Mg^{2+}$ supply by extraembryonic tissues, we used inductively coupled plasma mass spectrometry (ICP-MS) and found that relative magnesium ($Mg^{2+}$) levels were reduced in the whole e9.5 $Trpm6^{\beta geo/\beta geo}$ embryos (*Figure 1F*). Thus, *Trpm6* is specifically expressed in the placental labyrinth and the yolk sac at the stage when the $Mg^{2+}$ deficiency and growth delay of *Trpm6*-deficient embryos become apparent.

To investigate whether TRPM6 activity in extraembryonic cells underlies the lethality of *Trpm6* null embryos, we characterized a mouse strain with a 'floxed' ($Trpm6^{fl}$) allele (*Table 1*). Cre-mediated excision engendered viable mice heterozygous for the constitutive deletion mutation in *Trpm6* ($Trpm6^{\Delta 17/+}$). However, we were unable to produce live $Trpm6^{\beta geo/\Delta 17}$ or $Trpm6^{\Delta 17/\Delta 17}$ offspring, indicating that $Trpm6^{\Delta 17}$ is a true null mutation (*Table 1*). The paternally inherited *Sox2-Cre* trans-gene drives recombination only in epiblast cells, but not in extraembryonic tissues (*Hayashi et al.,*

**Table 1.** Postnatal survival of the mice with global and tissue-restricted deletions of *Trpm6*.

| Targeted tissue | Breeding strategy | Expected F1 outcome[*] | Survival of the mutant |
|---|---|---|---|
| Constitutive mutagenesis | | | |
| Whole fetus | ♂Trpm6$^{βgeo/+}$ x ♀Trpm6$^{βgeo/+}$ | 25%Trpm6$^{βgeo/βgeo}$<br>50% Trpm6$^{βgeo/+}$<br>25% Trpm6$^{+/+}$ | no |
| Whole fetus | ♂Trpm6$^{Δ17/+}$ x ♀Trpm6$^{Δ17/+}$ | 25% Trpm6$^{Δ17/Δ17}$<br>50% Trpm6$^{Δ17/+}$<br>25% Trpm6$^{+/+}$ | no |
| Whole fetus | ♂Trpm6$^{Δ17/+}$ x ♀Trpm6$^{βgeo/+}$ | 25% Trpm6$^{βgeo/Δ17}$<br>25% Trpm6$^{βgeo/+}$<br>25% Trpm6$^{Δ17/+}$<br>25% Trpm6$^{+/+}$ | no |
| Conditional mutagenesis using Cre/LoxP system | | | |
| Epiblast | ♂Trpm6$^{Δ17/+}$;Sox2-Cre x ♀Trpm6$^{fl/fl}$ | 25% Trpm6$^{Δ17/Δ17}$;Sox2-Cre<br>25% Trpm6$^{Δ17/fl}$<br>25% Trpm6$^{Δ17/+}$;Sox2-Cre<br>25% Trpm6$^{fl/+}$ | yes |
| Intestine | ♂Trpm6$^{Δ17/+}$;Villin1-Cre x ♀Trpm6$^{fl/fl}$ | 25% Trpm6$^{Δ17/fl}$;Villin1-Cre[†]<br>25% Trpm6$^{Δ17/fl}$<br>25% Trpm6$^{fl/+}$;Villin1-Cre<br>25% Trpm6$^{fl/+}$ | yes |
| Kidney | ♂Trpm6$^{Δ17/+}$;Ksp-Cre x ♀Trpm6$^{fl/fl}$ | 25% Trpm6$^{Δ17/fl}$;Ksp-Cre[†]<br>25% Trpm6$^{Δ17/fl}$<br>25% Trpm6$^{fl/+}$;Ksp-Cre<br>25% Trpm6$^{fl/+}$ | yes |

[*]Genotypes were determined using genomic DNA extracted from tail fragments.

[†]Individuals were homozygous for *Trpm6$^{Δ17}$* allele in the targeted cells.

2003). Notably, intercrosses of *Trpm6$^{Δ17/+}$;Sox2-Cre* males and *Trpm6$^{fl/fl}$* females resulted in viable *Trpm6$^{Δ17/Δ17}$* pups at the expected ratio (**Table 1**). Therefore, the embryonic mortality of *Trpm6*-deficient mice appears to be caused by the loss of TRPM6 in extraembryonic tissues.

## *Trpm6*-deficient adult mice display shortened lifespan, growth defects and Mg$^{2+}$ deficiency

We next studied the impact of a global deletion of *Trpm6* postnatally. Examination of *Trpm6*-deficient (*Trpm6$^{Δ17/Δ17}$;Sox2-Cre*) mice at weaning did not reveal conspicuous abnormalities. However, during the follow-up period, we observed the gradual development of pathologies. Thus, *Trpm6*-deficient mice had a lifespan of no longer than 16 weeks (**Figure 2A**). Mutants were growth-delayed, and displayed a lighter fur colour and low night-time activity (**Figure 2B,C**). Weight gain and lean body mass of *Trpm6*-deficient mice were reduced (**Figure 2D,E**), as was the muscle fibre area of the gastrocnemius muscle of 12–13 week-old *Trpm6*-deficient mice indicative of sarcopenia (**Figure 2F**). Mutant mice displayed kyphosis (**Figure 2G**) and completely lacked abdominal and subcutaneous fat depots (**Figure 2H,I**) indicative of catabolic metabolism. However, the total amount of faeces (**Figure 2—figure supplement 1A**) and the calorimetrically determined faecal energy content, as a measure of energy excretion (**Figure 2—figure supplement 1B**), were not altered in *Trpm6* mutants, ruling out insufficient food intake. Histological analysis of internal organs (**Figure 3**) showed that *Trpm6*-deficient mice developed lung emphysema and degeneration of lymphoid organs. Thus, the thymus of mutant mice was rudimentary and the cortex region was not distinguishable. In the spleen of *Trpm6*-deficient mice, the red pulp was substantially reduced. Hepatocytes of *Trpm6*-deficient mice were depleted of glycogen granules (**Figure 3**), corroborating catabolic metabolism. It has been suggested that low serum Mg$^{2+}$ and TRPM6 function are associated with atherosclerosis in humans (**Maier, 2012**; **Tin et al., 2015**). Therefore, we investigated whether such a phenotype would develop in our mouse model as well. However, examination of thoracal aorta showed no signs of atherosclerosis development in mutant mice (**Figure 2—figure supplement 2**).

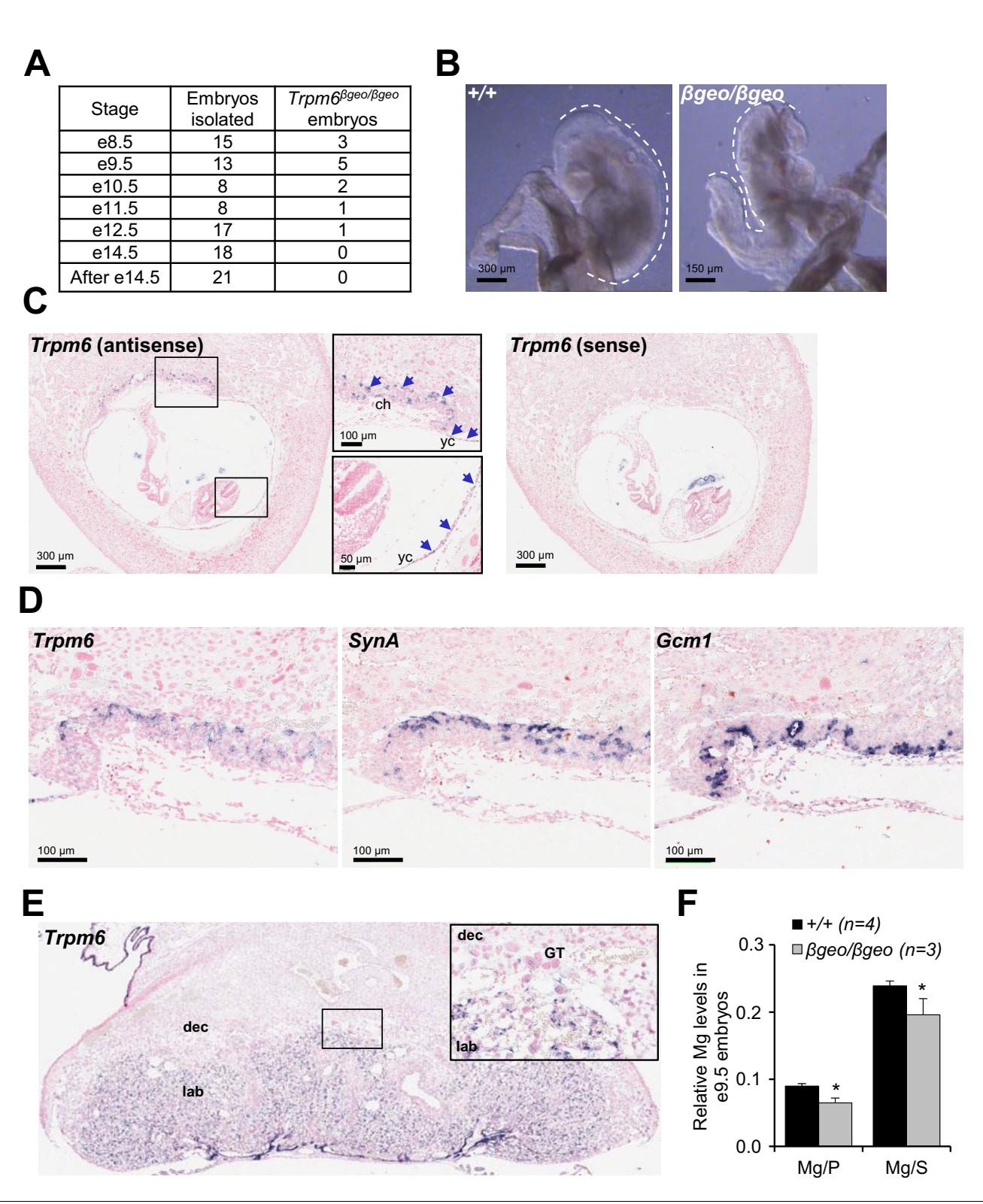

**Figure 1.** Assessment of *Trpm6* function in extraembryonic tissues. (**A**) Survival of *Trpm6^{βgeo/βgeo}* embryos obtained from *Trpm6^{βgeo/+}* intercrosses. (**B**) Representative images of e9.5 *Trpm6^{+/+}* (+/+, n = 13) and *Trpm6^{βgeo/βgeo}* (βgeo/βgeo, n = 5) embryos from dataset in (**A**). Dashed lines underline C-shaped versus S-shaped morphology of *Trpm6^{+/+}* and *Trpm6^{βgeo/βgeo}* embryos, respectively. (**C**) ISH on serial paraffin sections obtained from wildtype n = 5 e8.5 fetus using antisense (left) and sense (right) probes for *Trpm6*. Boxes indicate the positions of the magnified images of the chorion
*Figure 1 continued on next page*

Chubanov *et al.* eLife 2016;5:e20914. DOI: 10.7554/eLife.20914

Figure 1 continued

(*ch*) and yolk sac (*yc*). Arrows indicate *Trpm6*-positive cells in the developing labyrinth (chorion) and the endoderm layer in the visceral yolk sac. (**D**) ISH on serial paraffin sections of wildtype e8.5 placenta using DIG-labelled probes for *Trpm6* (*left*), *SynA* (*middle*) and *Gcm1* (*right*), respectively. *Note: Trpm6* expression was restricted to cells positive for *SynA*, a marker of SynT-I, and absent in cells expressing *Gcm1*, a marker of SynT-II. Representative images of n = 2 independent tissues are shown. (**E**) ISH of WT e14.5 placenta with the antisense *Trpm6* probe. The box indicates the position of the magnified image. The *Trpm6* signal is restricted to the labyrinth (*lab*) and not detectable in the decidua (*dec*) and trophoblast giant cells (GT). Representative images of n = 8 independent placentas are shown. (**F**) $Mg^{2+}$ levels in e9.5 $Trpm6^{+/+}$ (n = 4) and $Trpm6^{\beta geo/\beta geo}$ (n = 3) embryos. Distal segments of the embryos were used for genotyping, and the remaining parts were analysed by ICP-MS. Elementary magnesium (Mg) contents were normalized to phosphorus (P) and sulfur (S) levels represented as mean±SEM. *-$p \leq 0.05$ (Student's t-test).
The following figure supplement is available for figure 1:

**Figure supplement 1.** ISH on serial paraffin sections obtained from wildtype e8.5 fetus using antisense (left) and sense (right) probes for *Trpm6*.

Next, we asked whether the phenotype of *Trpm6*-deficient mice is caused by $Mg^{2+}$ deficiency. We employed ICP-MS to compare the concentrations of main elements in serum from controls and *Trpm6*-deficient littermates. We found that, similar to humans with mutations in the *TRPM6* gene (*Schlingmann et al., 2002*; *Walder et al., 2002*), *Trpm6*-deficient mice developed hypomagnesemia (*Figure 2J*). Serum $Mg^{2+}$ levels of mutant mice were only 0.58 mM (36% of control value, 1.58 mM), whereas concentrations of other elements were not changed (*Figure 2J*). A $Mg^{2+}$-enriched diet is an efficient way to alleviate hypomagnesemia in humans lacking TRPM6 (*Schlingmann et al., 2002*; *Walder et al., 2002*). Therefore, we asked whether the phenotypes of *Trpm6*-deficient mice were caused by $Mg^{2+}$ deprivation and could be rescued by dietary supplementation. To this end, we changed the regular chow (0.22% $Mg^{2+}$) of 4 week-old mutant mice and control littermates for a $Mg^{2+}$ enriched diet (0.75% $Mg^{2+}$). Notably, none of the *Trpm6*-deficient mice died during the following 12 weeks of $Mg^{2+}$ supplementation (*Figure 2K*). However, returning to regular chow resulted in 100% mortality of mutant mice within the following 13 weeks (*Figure 2K*). $Mg^{2+}$ supplemented mutants neither exhibited kyphosis nor lipodystrophy (data not shown). Furthermore, the morphology of the lung, spleen, and thymus of $Mg^{2+}$ supplemented mutants closely resembled that of control mice (*Figure 3*). We asked whether dietary $Mg^{2+}$ supplementation of $Trpm6^{\beta geo/+}$ parents would benefit the survival of *Trpm6*-deficient offspring. However, similar to a previous study (*Walder et al., 2009*), we found that this treatment was inefficient.

*Trpm6*-deficient adult mice phenocopy salient pathologies reported for a set of mouse strains advocated as genetic models of 'accelerated' or 'premature' aging (*Kuro-o et al., 1997*; *Trifunovic et al., 2004*; *Kujoth et al., 2005*; *Varela et al., 2005*; *Mostoslavsky et al., 2006*; *Niedernhofer et al., 2006*; *van der Pluijm et al., 2007*; *López-Otín et al., 2013*). Similar to *Trpm6*-deficient mice, the latter mutants display short lifespan, growth failure, low physical activity, kyphosis, lung emphysema, sarcopenia, lipodystrophy and degeneration of lymphoid organs. A characteristic feature of these mouse strains is suppression of the somatotropic axis accompanied by induction of xenobiotic detoxification gene networks in the liver (*Niedernhofer et al., 2006*; *van de Ven et al., 2006*; *van der Pluijm et al., 2007*; *Schumacher et al., 2008*; *Garinis et al., 2009*; *Mariño et al., 2010*), interpreted as a defensive organismal response, slowing down growth and metabolism in favor of somatic preservation (*López-Otín et al., 2013*). We asked whether *Trpm6*-deficient mice would also display such protective metabolic responses. In fact, we found that serum IGF1 concentrations were reduced in *Trpm6*-deficient mice as well (*Figure 4A*). Mutant mice had a lower core body temperature (*Figure 4B*) and a profoundly reduced urinary content of major urinary proteins (MUPs) (*Figure 4C*), two known features of supressed IGF1 signalling (*Mariño et al., 2010*; *Bartke et al., 2013*). Even though mutant mice showed signs of overall energy shortage, circulating levels of ketone bodies (ß-hydroxybutyrate) were not elevated (*Figure 2—figure supplement 1C*). When subjected to an oral glucose tolerance test, mutant mice displayed lower peripheral glucose concentrations than controls despite of a similar amount of insulin released, thus reflecting increased insulin sensitivity (*Figure 2—figure supplement 1D,E*), another hallmark of suppressed IGF1 signalling (*Bartke et al., 2013*). Notably, body weight and IGF1 serum levels were indistinguishable in $Mg^{2+}$ supplemented mutant and control mice suggesting that the variations observed in mice maintained on a regular diet were induced by $Mg^{2+}$ deficiency (*Figure 2—figure supplement 1F,G*).

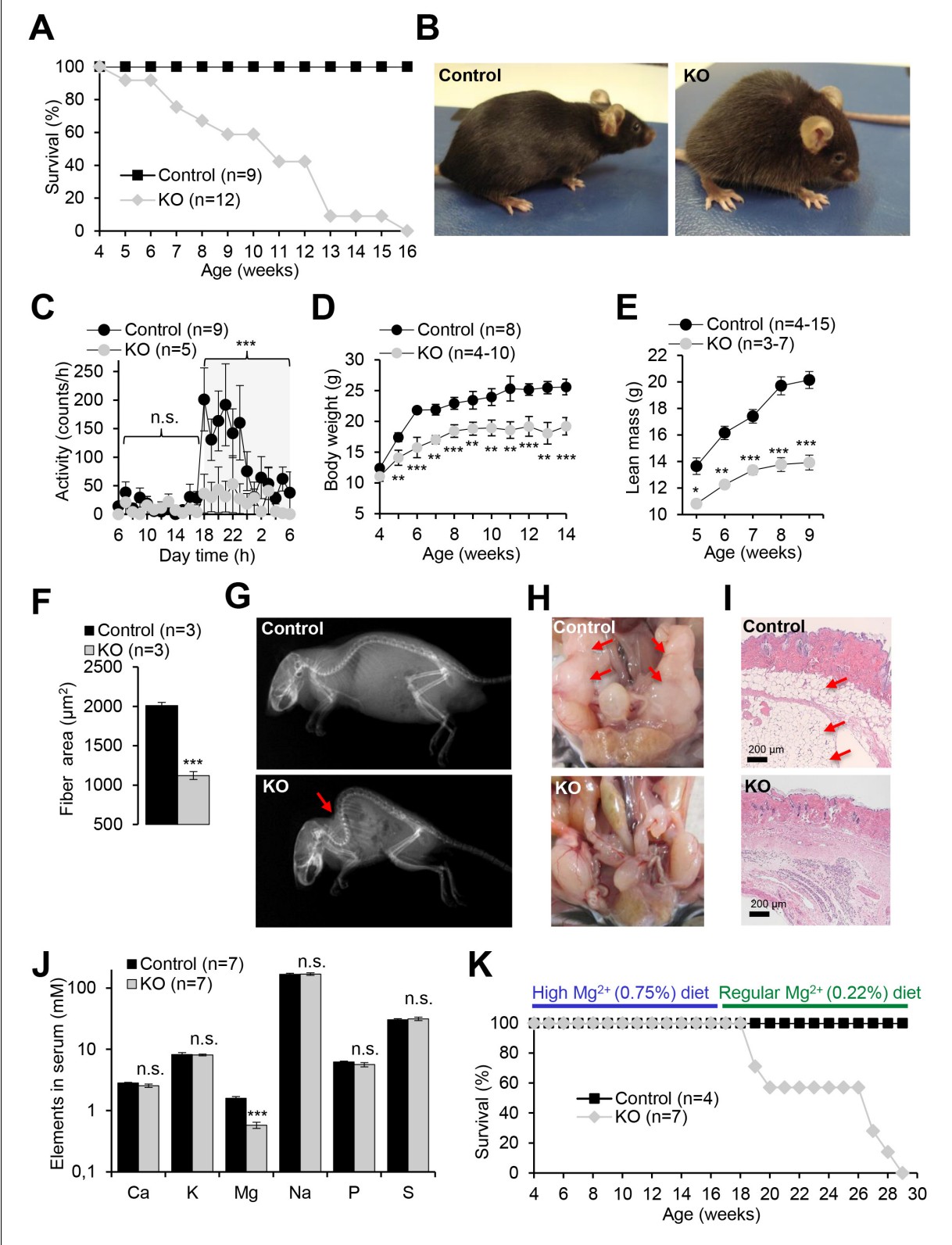

**Figure 2.** Pathophysiological changes displayed by *Trpm6*-deficient adult mice. Unless stated otherwise, 10–12 week-old *Trpm6*$^{fl/+}$ (*Control*) and *Trpm6*$^{\Delta17/\Delta17}$;*Sox2-Cre* (*KO*) littermates were studied. (A–E) Mice were examined for survival rate (A), overall physical appearance (B), day/night activity of 8 week-old individuals (C), growth rate (D) and lean mass (E). (F) Fibre size of the gastrocnemius muscle after hematoxylin-eosin staining. (G) X-ray images of mice. The red arrow indicates the characteristic skeletal deformation (kyphosis) observed in *Trpm6*-deficient mice. (H) Assessment of

*Figure 2 continued on next page*

*Figure 2 continued*

abdominal fat. Arrows indicate fat deposits observed only in control mice. (I) H and E staining of paraffin skin sections. Arrows indicate a layer of fat cells present only in control mice. Histological analysis was performed with three animals per group resulting in similar observations. (J) The levels of main elements in the serum of 8 week-old mice assessed by ICP-MS. (K) The survival rate of mice maintained on high $Mg^{2+}$ (0.75%) and regular (0.22%) chows. Data are represented as mean±SEM. \*\*\*-p≤0.001; \*\*-p≤0.01; \*-p≤0.05; n.s. – not significantly different (Student's t-test); n – number of mice examined.

The following figure supplements are available for figure 2:

**Figure supplement 1.** Examination of energy balance in *Trpm6*-deficient mice.
**Figure supplement 2.** Evaluation of atherosclerosis development in *Trpm6*-deficient mice.

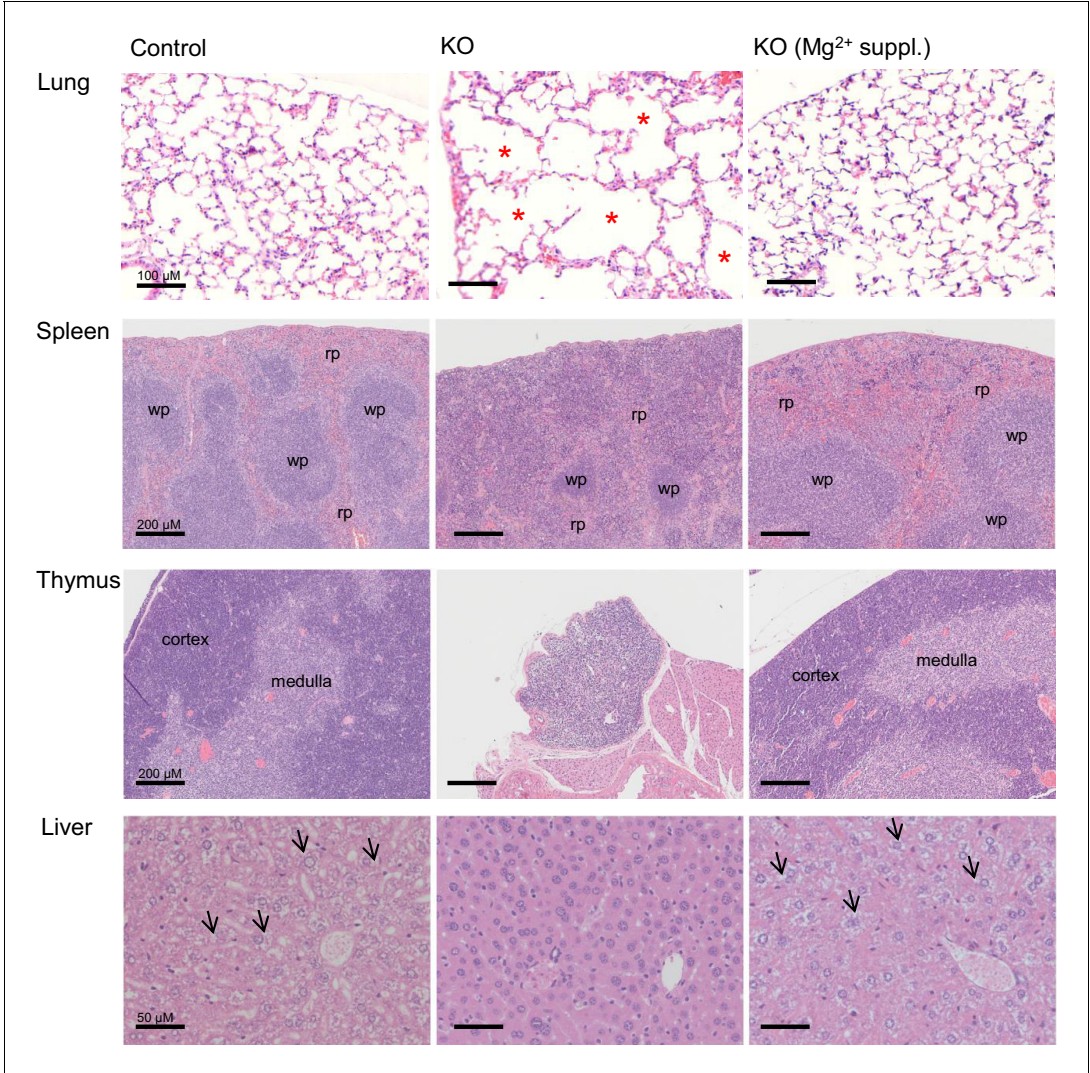

**Figure 3.** Histology of internal organs of *Trpm6*-deficient mice. Hematoxylin-eosin staining of paraffin embedded tissue sections of 12–13 week-old control (*Control*) and *Trpm6*-deficient (*KO*) mice maintained either on regular (0.22% $Mg^{2+}$) or $Mg^{2+}$ supplemented (0.75% $Mg^{2+}$) chows. *Trpm6*-deficient mice maintained on the regular diet showed marked airspace enlargement (indicated by stars) mimicking lung emphysema, distortion of splenic red pulp (*rp*)/ white pulp (*wp*) microarchitecture, thymic atrophy, and reduction of intracellular glycogen in hepatocytes (indicated by arrows). Histological analysis was performed with three animals per group resulting in similar observations.

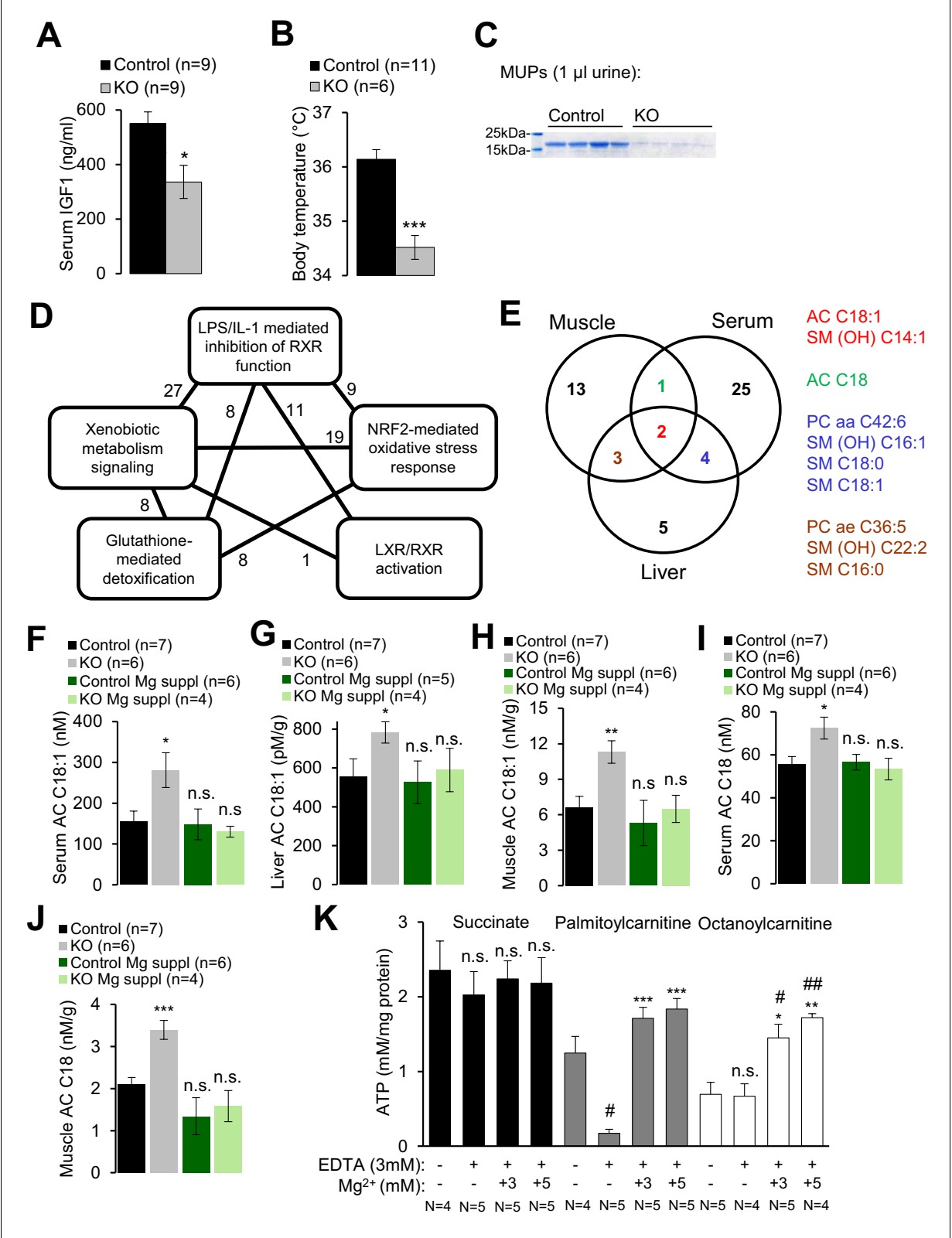

**Figure 4.** Assessment of metabolic profiles of *Trpm6*-deficient mice. (**A–C**) 8 week-old *Trpm6*[fl/+] (*Control*) and *Trpm6*[Δ17/Δ17];*Sox2-Cre* (*KO*) littermates were evaluated for (**A**) serum IGF1, (**B**) body temperature, (**C**) urinary MUPs content in individual mice. Data are represented as mean±SEM. ***-p≤0.001; *-p≤0.05 (Student's t-test); n – number of mice examined. (**D**) IPA analysis of genome-wide hepatic transcriptome profiling of *Trpm6*-deficient (n = 3) *vs* control (n = 4) littermates. The diagram shows the top 5 of IPA Canonical Pathways significantly changed in mutant mice (***Supplementary file***

*Figure 4 continued*

2). Numbers of the commonly changed transcripts are indicated close to the lines connecting the pathways. (E) Venn diagram for sets of metabolites significantly changed (FDR p≤0.05) in serum, liver and gastrocnemius muscle *Trpm6*-deficient (n = 6) *vs* control (n = 8) littermates (*Supplementary file 3*). Commonly changed metabolites are listed in different colours as outlined in the Venn diagram. (F–J) Levels of AC C18:1 (F–H) and AC C18 (I–J) examined in the serum (F, I), liver (G) and gastrocnemius muscle (H, J) of *Trpm6*-deficient and control mice. Data are represented as mean±SEM. ***-p≤0.001; **-p≤0.01; *-p≤0.05; n.s. – not significantly different to control group maintained on a regular $Mg^{2+}$ diet (one-way ANOVA); n – number of mice examined. (K) ATP production by mitochondria isolated from the liver of wildtype C57BL/6 mice with succinate, palmitoylcarnitine or octanoylcarnitine as energy sources. ATP levels were determined after 30 min incubation of untreated (-) or treated (+) mitochondria by EDTA with or without $Mg^{2+}$. Data are represented as mean±SEM of 4–5 independent isolations (N). ##-p≤0.01; #-p≤0.05 significantly different to the control group; ***-p≤0.001; **-p≤0.01; *-p≤0.05 significantly different to the EDTA treated group (Student's t-test). n.s. – not significantly different.

The following figure supplements are available for figure 4:

**Figure supplement 1.** Gene expression profiling of *Trpm6*-deficient and control mice.

**Figure supplement 2.** Metabolomic profiling of *Trpm6*-deficient and control mice.

**Figure supplement 3.** Assessment of the membrane potential ($\Delta\psi_m$) in isolated mitochondria.

Finally, we performed whole-genome profiling of the liver transcriptome (*Supplementary file 1*). Applying a cut-off value of p≤0.1 for the false discovery rate (FDR), we identified 46 genes up- or down-regulated in the livers of *Trpm6*-deficient mice (*Figure 4—figure supplement 1*, *Supplementary file 1*). The majority of affected transcripts code for solute carrier transporters, cytochrome P450 metabolising enzymes, glutathione S-transferases and proteins metabolizing steroids. As expected, Ingenuity Pathway Analysis (IPA) revealed that the inactivation of *Trpm6* is associated with an induction of interconnected gene networks controlling toxicity responses and xenobiotic metabolism governed by nuclear receptors such as retinoid X receptors (RXR), liver X receptor (LXR) and farnesoid X receptor (FXR) (*Figure 4D*, *Supplementary file 2*). Hence, *Trpm6*-deficient mice were characterized by suppression of the somatotropic axis and induction of xenobiotic detoxification responses.

### *Trpm6*-deficient mice display insufficient utilization of long-chain acylcarnitines

To gain mechanistic insight into the altered energy metabolism of mutant mice, we quantified serum, liver and skeletal muscle levels of 237 metabolites (*Supplementary file 3*, *Figure 4—figure supplement 2*). Unexpectedly, alterations in *Trpm6*-deficient mice (FDR p≤0.1) were restricted to only several metabolites representing mainly long-chain acylcarnitines (AC), phosphatidylcholines (PC), and sphingomyelins (SM) (*Figure 4E*). These findings, as well as the results of gene array profiling (*Figure 4D*) suggest that sustained $Mg^{2+}$ deficiency triggers a specific metabolic response rather than widespread unsystematic changes. In line with this idea, we noted that AC C18:1 and the related AC C18 were consistently increased in tissues of *Trpm6*-deficient mice (*Figure 4F–J*), whereas carnitine levels were not changed (*Supplementary file 3*). In conjunction with lowered concentrations of glucose and unchanged ketogenesis (*Figure 2—figure supplement 1C*), this constellation is a metabolic 'signature' of a frequent inherited human disorder characterized by inefficient β-oxidation of fatty acids due to mitochondrial carnitine palmitoyltransferase II deficiency (*Gempel et al., 2002*; *Bonnefont et al., 2004*). AC C18:1 and C18 levels were normalized in serum and tissues obtained from $Mg^{2+}$ supplemented mutants (*Figure 4F–J*), implying that metabolic changes of AC were caused by $Mg^{2+}$ deficiency.

Consequently, we asked whether $Mg^{2+}$ would specifically affect mitochondrial ATP production (*Figure 4K*) and maintenance of the mitochondrial membrane potential (MMP) (*Figure 4—figure supplement 3A*). To address this question, wildtype liver mitochondria were incubated in a buffer containing 3 mM EDTA with or without 1–5 mM $Mg^{2+}$. Such manipulations had no negative effect on mitochondrial respiration when succinate was offered as an energy source (*Figure 4K*, *Figure 4—figure supplement 3A*). In contrast, mitochondria failed to utilize octanoylcarnitine or palmitylcarnitine for ATP production (*Figure 4K*) and to maintain MMP (*Figure 4—figure supplement 3A*) in the

presence of EDTA. Importantly, ATP production and MMP could be fully rescued by the administration of 3–5 mM $Mg^{2+}$ (*Figure 4K*, *Figure 4—figure supplement 3A*), but not $Zn^{2+}$ or $Ca^{2+}$ (*Figure 4—figure supplement 3B*). Thus, sustained $Mg^{2+}$ deprivation impairs energy homeostasis resulting in a catabolic metabolism in *Trpm6*-deficient mice, at least partially due to insufficient mitochondrial utilization of AC.

## *Trpm6* null mice develop $Mg^{2+}$ deficiency due to a defect in intestinal $Mg^{2+}$ uptake

Next, we investigated the etiology of hypomagnesemia in *Trpm6*-deficient mice. ~50% of body $Mg^{2+}$ content is stored in bones, ~30% in muscle tissues and only ~1% in the serum (*de Baaij et al., 2015*). Using ICP-MS we studied the $Mg^{2+}$ content in bones (right tibia) and gastrocnemius muscle, and observed that $Mg^{2+}$ levels in bones of *Trpm6* null mice were only 24% of control values (*Figure 5A*). Furthermore, the $Mg^{2+}$ content of muscle was also significantly reduced in *Trpm6*-deficient mice (*Figure 5B*). Hence, *Trpm6*-deficient mice develop a severe systemic $Mg^{2+}$ deficit.

It is generally assumed that the distal convoluted tubule (DCT) of the kidney is critical for whole-body $Mg^{2+}$ balance (*de Baaij et al., 2015*). Accordingly, immunostaining of control kidney cryosections with a TRPM6-specific antibody labelled nephron segments resembling DCT (*Figure 5C*). TRPM6 was not detectable in the kidneys of *Trpm6*-deficient mice. Surprisingly, mutant mice exhibited substantially reduced urinary $Mg^{2+}$ excretion (only 14% of control values, *Figure 5D*), whereas fecal $Mg^{2+}$ loss was significantly increased (166%, *Figure 5E*), indicating that mice lacking TRPM6 develop $Mg^{2+}$ deficiency primarily due to impaired intestinal $Mg^{2+}$ uptake. Therefore, we studied the expression pattern of *Trpm6* in the intestine. Because the TRPM6-specific antibody did not efficiently and specifically detect TRPM6 protein in the intestine, we resorted to ISH (*Figure 5—figure supplement 1*). *Trpm6* transcripts were not detectable in the small intestine, but *Trpm6*-specific signal was observed in absorptive epithelial cells of the colon (*Figure 5—figure supplement 1*). The colon of mutant mice was not stained by a *Trpm6*-specific ISH probe (*Figure 5F*), consistent with the notion that systemic $Mg^{2+}$ deficit in *Trpm6*-deficient mice was primarily caused by a defect of $Mg^{2+}$ uptake in the colon.

To directly assess the contribution of the kidney versus intestine to the *Trpm6* null phenotype, we employed *Ksp-Cre* (*Shao et al., 2002*) and *Villin1-Cre* (*Madison et al., 2002*) transgenic mice to specifically ablate floxed *Trpm6* alleles in renal and intestinal epithelial cells, respectively (*Table 1*). Surprisingly, conditional *Trpm6* inactivation in the kidney neither impacted serum $Mg^{2+}$ concentration nor urinary $Mg^{2+}$ excretion, and bone $Mg^{2+}$ content was only slightly reduced (*Figure 5G–J*). In contrast, disruption of *Trpm6* in the intestine resulted in hypomagnesemia, reduced bone $Mg^{2+}$ content and lowered urinary $Mg^{2+}$ excretion (*Figure 5K–N*), indicating that wildtype kidneys are not able to compensate for the ablation of intestinal TRPM6. Hence, in contrast to current thinking, our findings support the new concept that *Trpm6*-dependent $Mg^{2+}$ uptake in the intestine plays an indispensable role for systemic $Mg^{2+}$ balance.

## TRPM6 cooperates with TRPM7 to regulate divalent cation currents

TRPM6 is invariably co-expressed with the TRPM7 channel and the question as to why TRPM6 function is non-redundant remains central to a mechanistic understanding of the *Trpm6* null phenotype. Because *Trpm6* expression levels in the epithelial cells of the colon (*Figure 5—figure supplement 1*) and the kidney (*Figure 5C*) are highly heterogeneous, we searched for an alternative native cell model to dissect the functional interplay of TRPM6 and TRPM7. Trophoblast stem (TS) cells are widely used to study the transport function of placental trophoblasts (*Tanaka et al., 1998*; *Simmons and Cross, 2005*) and, as shown before (*Figure 1*), this cell type is crucial for the fetal *Trpm6* phenotype. Therefore, we derived *Trpm6*$^{+/+}$ and *Trpm6*-deficient (*Trpm6*$^{\beta geo/\beta geo}$) TS cells from e3.5 blastocysts isolated from *Trpm6*$^{\beta geo/+}$ parents (*Figure 6—figure supplement 1A*). We also produced *Trpm7*$^{+/+}$ and *Trpm7*-deficient (*Trpm7*$^{\Delta 17/\Delta 17}$) TS cells (*Figure 6—figure supplement 2A*) using *Trpm7*$^{\Delta 17/+}$ mice (*Jin et al., 2008*). As expected, RT-PCR analysis revealed that wildtype TS cells expressed TRPM6 and TRPM7 (*Figure 6—figure supplement 1B*, *Figure 6—figure supplement 2B*). *Trpm6*$^{\beta geo/\beta geo}$ TS cells were maintained in culture for >40 passages. Furthermore, analysis of DNA content showed that the proportion of polyploidy was similar in *Trpm6*$^{+/+}$ and *Trpm6*$^{\beta geo/\beta geo}$ TS cells (*Figure 6—figure supplement 1C,D*), suggesting that inactivation of *Trpm6*

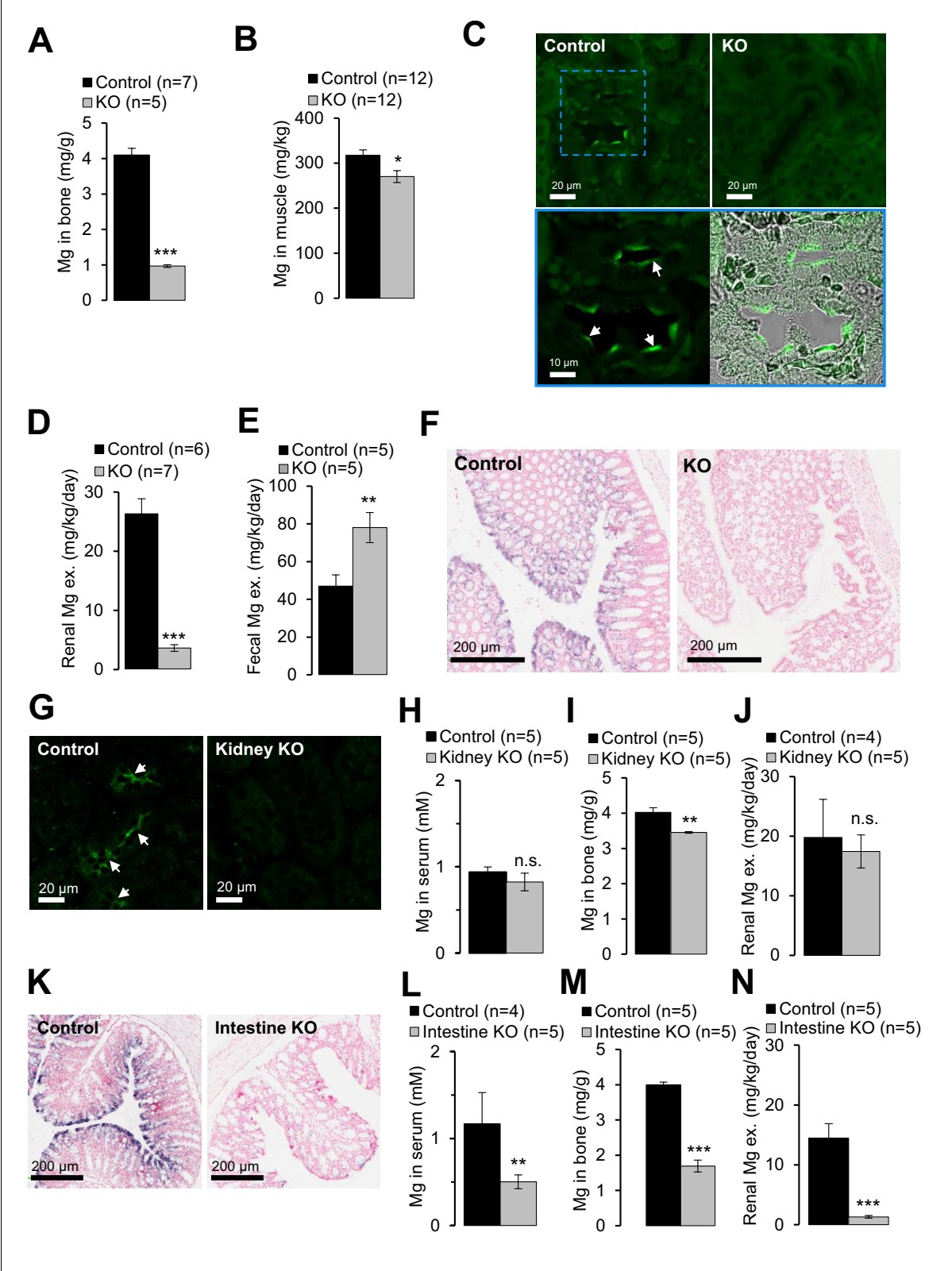

**Figure 5.** Examining of Mg²⁺ balance in *Trpm6*-deficient adult mice. (A–F) Assessment of 8 week-old *Trpm6^fl/+* (*Control*) and *Trpm6^Δ17/Δ17;Sox2-Cre* (*KO*) littermate males. (A) Mg²⁺ levels in bones and (B) gastrocnemius muscle assessed by ICP-MS. (C) Immunostaining of kidney cryosections using a TRPM6-specific antibody. Representative images are shown (n = 2 tissues per genotype). The blue square indicates the position of the confocal and differential interference contrast magnified images acquired from control tissue. Arrows indicate labelling of the apical surface of renal tubules. (D) 24 hr

*Figure 5 continued on next page*

Figure 5 continued

urinary and (E) fecal $Mg^{2+}$ excretion rates. (F) ISH on paraffin sections obtained from the colon of control and *Trpm6*-deficient mice (n = 2 tissues per genotype). (G–J) Examination of 6 month-old *Trpm6*$^{fl/+}$ (*Control*) and *Trpm6*$^{\Delta 17/fl}$;*Ksp-Cre* (*Kidney KO*) littermate males. (G) Immunostaining of TRPM6 in kidney cryosections. *Arrows* indicate labelling of renal tubules. (H–I) Determination of $Mg^{2+}$ in serum (H) and bones (I). (J) 24 hr urinary $Mg^{2+}$ excretion rate. (K–N) Assessment of 6 month-old *Trpm6*$^{fl/+}$ (*Control*) and *Trpm6*$^{\Delta 17/fl}$;*Villin1-Cre* (*Intestine KO*) littermate males. (K) ISH on paraffin sections of the colon using a *Trpm6*-specific probe (n = 2 tissues per genotype). (L, M) $Mg^{2+}$ levels in the serum (L) and bones (M). (N) 24 hr urinary $Mg^{2+}$ excretion rate. Data are represented as mean±SEM. ***-$p \leq 0.001$; **-$p \leq 0.01$; *-$p \leq 0.05$; n.s. – not significantly different (Student's t-test); n – number of mice examined. Histological analysis in (F) and (K) was performed with n = 3 animals per group resulting in similar observations.

The following figure supplement is available for figure 5:

**Figure supplement 1.** Expression pattern of *Trpm6* in the intestine.

did not affect the self-renewal of *Trpm6*$^{\beta geo/\beta geo}$ stem cells. In contrast, *Trpm7*$^{\Delta 17/\Delta 17}$ TS cells did not proliferate, unless the cell culture medium was supplemented with additional $Mg^{2+}$ (*Figure 6—figure supplement 2C*), supporting the concept that TRPM7 plays a pivotal role in cellular $Mg^{2+}$ uptake that cannot be maintained by TRPM6 alone (*Schmitz et al., 2003*; *Chubanov et al., 2004*; *Ryazanova et al., 2010*).

TRPM6 and TRPM7 have been suggested as molecular correlates of MgATP- and $Mg^{2+}$-regulated cation currents responsible for the cellular uptake of divalent cations including $Mg^{2+}$ (*Aarts et al., 2003*; *Nadler et al., 2001*; *Schmitz et al., 2003*; *Chubanov et al., 2004*; *Voets et al., 2004*; *Zhang et al., 2014*). Patch-clamp analysis showed that *Trpm6*$^{\beta geo/\beta geo}$ TS cells display substantially reduced TRPM6/M7-like outward currents at +80 mV (*Figure 6A*). Due to permeation block by extracellular divalent cations (*Nadler et al., 2001*; *Fleig and Chubanov, 2014*), inward currents at physiological membrane potentials were very small (*Figure 6A*; $p \leq 0.001$, t-test). However, exposure of TS cells to a divalent cation free (DVF) solution resulted in large monovalent cation currents (*Figure 6B*). Under these conditions, mutant TS cells exhibited a comparable reduction of inward ($p \leq 0.01$, t-test) as well as outward ($p \leq 0.05$, t-test) monovalent currents (*Figure 6B*).

Cytosolic levels of free $Mg^{2+}$ ($[Mg^{2+}]_i$) and MgATP ($[MgATP]_i$) have been suggested to exert a negative feedback mechanism on TRPM6/M7 channel activity (*Fleig and Chubanov, 2014*). We observed (*Figure 6C*) that currents in *Trpm6*$^{\beta geo/\beta geo}$ TS cells were more susceptible to concentration-dependent inhibition by cytosolic MgATP ($p \leq 0.0001$, F-test). The calculated $IC_{50}$ value for *Trpm6*$^{+/+}$ was 3.17 mM (Hill (h) slope = $-1.81$). Currents in *Trpm6*$^{\beta geo/\beta geo}$ cells were inhibited by $[MgATP]_i$ with an $IC_{50}$ value of 1.45 mM (h = $-1.95$; $p \leq 0.0015$, F-test). These results suggest that physiological concentrations of $[MgATP]_i$ varying between 2–7 mM in mammalian cells (*Günther, 2006*; *Romani, 2011*) will affect currents in *Trpm6*$^{+/+}$ and *Trpm6*$^{\beta geo/\beta geo}$ cells differently. In contrast, we observed no significant differences in $[Mg^{2+}]_i$ concentration-response curves for *Trpm6*$^{+/+}$ and *Trpm6*$^{\beta geo/\beta geo}$ currents (p=0.62, F-test) (*Figure 6D*). The obtained $IC_{50}$ value for *Trpm6*$^{+/+}$ currents was 0.60 mM (h = $-1.14$) and was not significantly altered in *Trpm6*$^{\beta geo/\beta geo}$ TS cells (0.72 mM, h = $-1.18$; p=0.22, F-test), suggesting that physiological concentrations (0.3–1 mM, (*Günther, 2006*; *Romani, 2011*)) of $[Mg^{2+}]_i$ will exert similar effects on ion currents in *Trpm6*$^{+/+}$ and *Trpm6*$^{\beta geo/\beta geo}$ cells.

Next, we asked whether TRPM6 channel activity would be detectable in the absence of TRPM7. Remarkably, we observed that *Trpm7*$^{\Delta 17/\Delta 17}$ TS cells completely lacked TRPM6/M7-like currents (*Figure 6E,F*). These findings cogently support our model (*Chubanov et al., 2004*) that native TRPM6 primarily functions in close cooperation with TRPM7. TRPM6 facilitates $Mg^{2+}$ uptake by increasing the amplitude of TRPM7-like currents and relieving TRPM7 from the negative feedback by MgATP (*Figure 6G*).

Finally, we studied whether $Mg^{2+}$ deprivation may affect energy metabolism at the cellular level. We chose the genetically tractable human haploid leukaemia cell line (HAP1 cells) (*Essletzbichler et al., 2014*; *Blomen et al., 2015*; *Wang et al., 2015*) as a new model system. CRISPR/Cas9-mediated ablation of the TRPM7 protein in HAP1 cells (*Figure 6—figure supplement 3A,B*) completely abolished TRPM7-like currents (*Figure 6—figure supplement 3C*). When cultured in standard medium for 24 hr, *TRPM7*-deficient cells were characterized by a reduced total cellular $Mg^{2+}$ content and a $Mg^{2+}$-dependent proliferation defect (*Figure 6—figure supplement 3D,E*). In

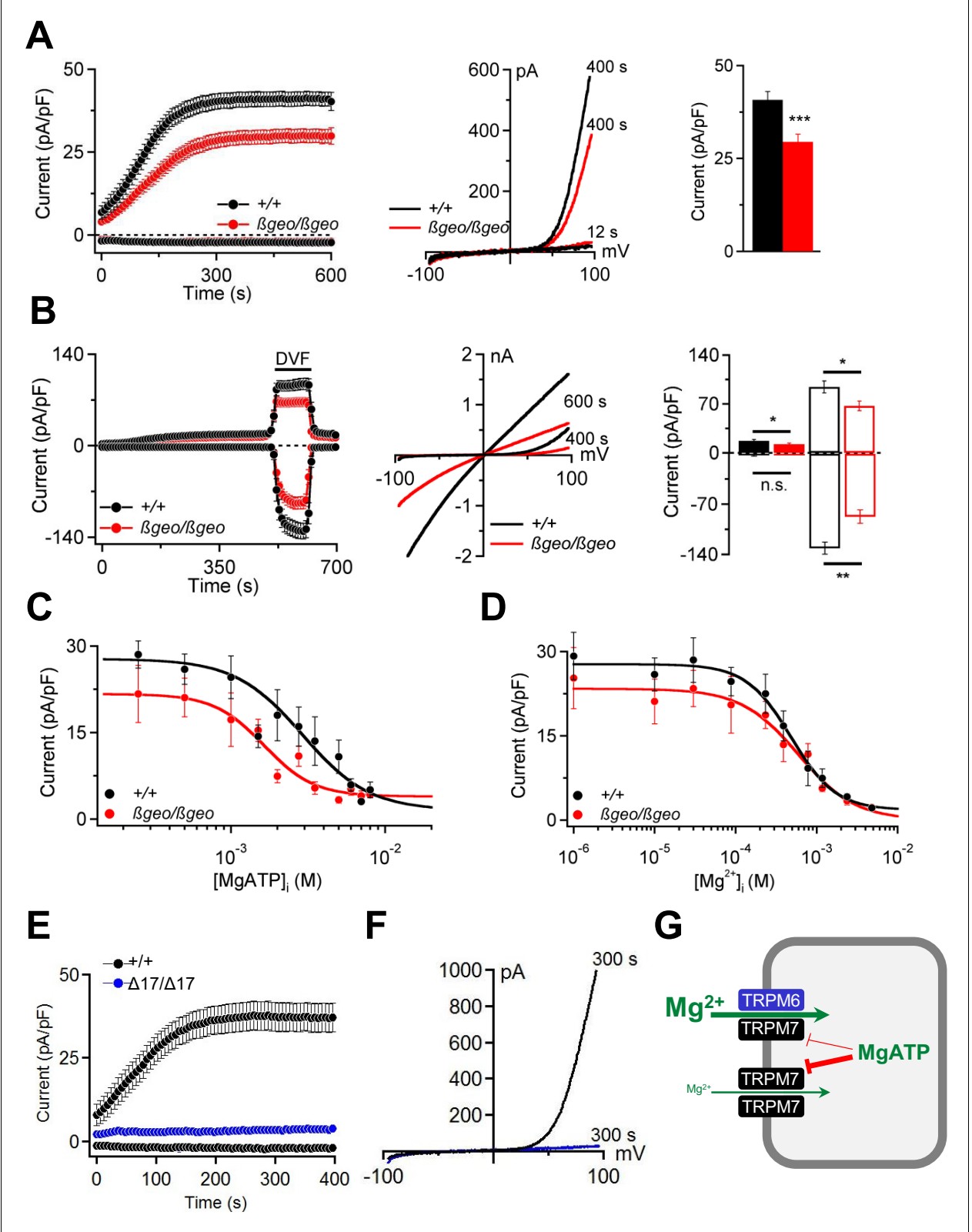

**Figure 6.** Characterization of TRPM6/M7-like currents in *Trpm6-* and *Trpm7*-deficient TS cells. (**A**) *Left panel:* Whole-cell currents measured at −80 mV and +80 mV over time in *Trpm6$^{+/+}$* (n = 22) and *Trpm6$^{ßgeo/ßgeo}$* (n = 22) TS cells. *Middle panel:* Representative current-voltage relationships obtained at 12 s and 400 s. *Right panel*: Bar graphs of current amplitudes at +80 mV (400 s). (**B**) Measurements were performed in control (n = 16) and *Trpm6*-deficient (n = 14) TS cells analogous to (**A**) except that the external saline (containing 2 mM Mg$^{2+}$ and 1 mM Ca$^{2+}$) was exchanged with divalent-free

*Figure 6 continued on next page*

*Figure 6 continued*

(*DVF*) solution (black bar). *Right panel* shows currents measured before (filled bars) and after application of DVF solution (open bars) at 400 s and 600 s, respectively. (**C, D**) Dose-dependent inhibition of currents (+80 mV, 400 s) by [MgATP]$_i$ and [Mg$^{2+}$]$_i$, respectively (n = 4–18 cells per concentration). (**E, F**) Whole-cell currents of $Trpm7^{+/+}$ (n = 15) and $Trpm7^{\Delta 17/\Delta 17}$ (n = 10) TS cells studied similar to (**A, B**). Data are represented as mean±SEM. ***-p≤0.001; **-p≤0.01; *-p≤0.05; n.s. – not significantly different (Student's t-test). n – number of cells examined. (**G**) A suggested model for the molecular role of TRPM6 in epithelial cells.

The following figure supplements are available for figure 6:

**Figure supplement 1.** Characterization of *Trpm6*-deficient TS cells.

**Figure supplement 2.** Examination of TS cells deficient in *Trpm7*.

**Figure supplement 3.** Evaluation of human haploid leukaemia (HAP) cells deficient in *TRPM7*.

addition, *TRPM7*-deficient HAP1 cells had reduced intracellular ATP levels (*Figure 6—figure supplement 3F*). Finally, the respiration rate of *TRPM7*-deficient HAP1 cells was significantly lower when compared to control cells (*Figure 6—figure supplement 3G,H*).

## Assessment of dietary Mg$^{2+}$ fortification on the lifespan of wildtype mice

Our studies with *Trpm6*-deficient mice clearly demonstrated that a sustained disruption of Mg$^{2+}$ homeostasis is detrimental for overall health and eventually reduces the lifespan of affected animals. Conversely, we asked whether a Mg$^{2+}$-enriched diet might exert a beneficial effect on the lifespan of wildtype mice. Since there are no prior reports on lifespan extension of mice subjected to life-long dietary Mg$^{2+}$ supplementation (or any other mineral), we performed proof-of-principle experiments to investigate, if such an effect can be observed. To this end, we used only the long-lived B6C3F1 hybrid mouse strain to avoid genotype-specific effects on disease susceptibility observed in longevity experiments with inbred strains (*Lipman et al., 1999*; *Turturro et al., 1999*; *Mitchell et al., 2016*). At this stage, we studied only females because of a larger number of early losses of males due to fighting (*Miller et al., 2007*). Finally, we studied animals only under pathogen-free conditions, since Mg$^{2+}$ may elicit a protective effect via the immune system (*Brandao et al., 2013*; *Chaigne-Delalande et al., 2013*). Consistent with published reports (*Turturro et al., 1999*), the mean lifespan of B6C3F1 mice (886 days regarded as 100%) was significantly extended (1100 days, 125%) by dietary restriction (DR) (*Figure 7A,B*, *Table 2*). Remarkably, supplementation with three Mg$^{2+}$ salts (Mg(CH$_3$COO)$_2$, Mg(OH)$_2$ and MgCl$_2$) increased the mean lifespan of mice by approximately 10% (976, 976 and 961 days, respectively). The nutritional CaCl$_2$ administration was without any significant effect (828 days). In contrast to DR, animals supplemented with Mg$^{2+}$ had a normal or even increased body weight (*Figure 7C*), ruling out the possibility that high dietary Mg$^{2+}$ affected the lifespan of mice due to reduced food intake. Hence, opposite to *Trpm6*-dependent Mg$^{2+}$ deprivation, dietary Mg$^{2+}$ supplementation may have a beneficial effect on the lifespan of mice suggestive future large-scale longevity studies with varied conditions and species.

## Discussion

Here, we present a new mechanistic model of the regulation of Mg$^{2+}$ homeostasis during development and postnatal life of mice. Against current thinking we show in vivo that TRPM6 is not required for embryonic development per se, but primarily operates in placenta and intestine to regulate Mg$^{2+}$ levels by transcellular transport, while TRPM6 function in the kidney – commonly thought to be essential – is expendable for organismal Mg$^{2+}$ balance. We demonstrate that ablation of TRPM6 in adult mice leads to reduced lifespan, growth defects and profoundly impaired health of mutant mice due to defective energy metabolism. We also show that dietary Mg$^{2+}$ supplementation is not only sufficient to prevent all *Trpm6* null pathologies, but Mg$^{2+}$ is the only mineral known so far able to extend the lifespan of wildtype mice.

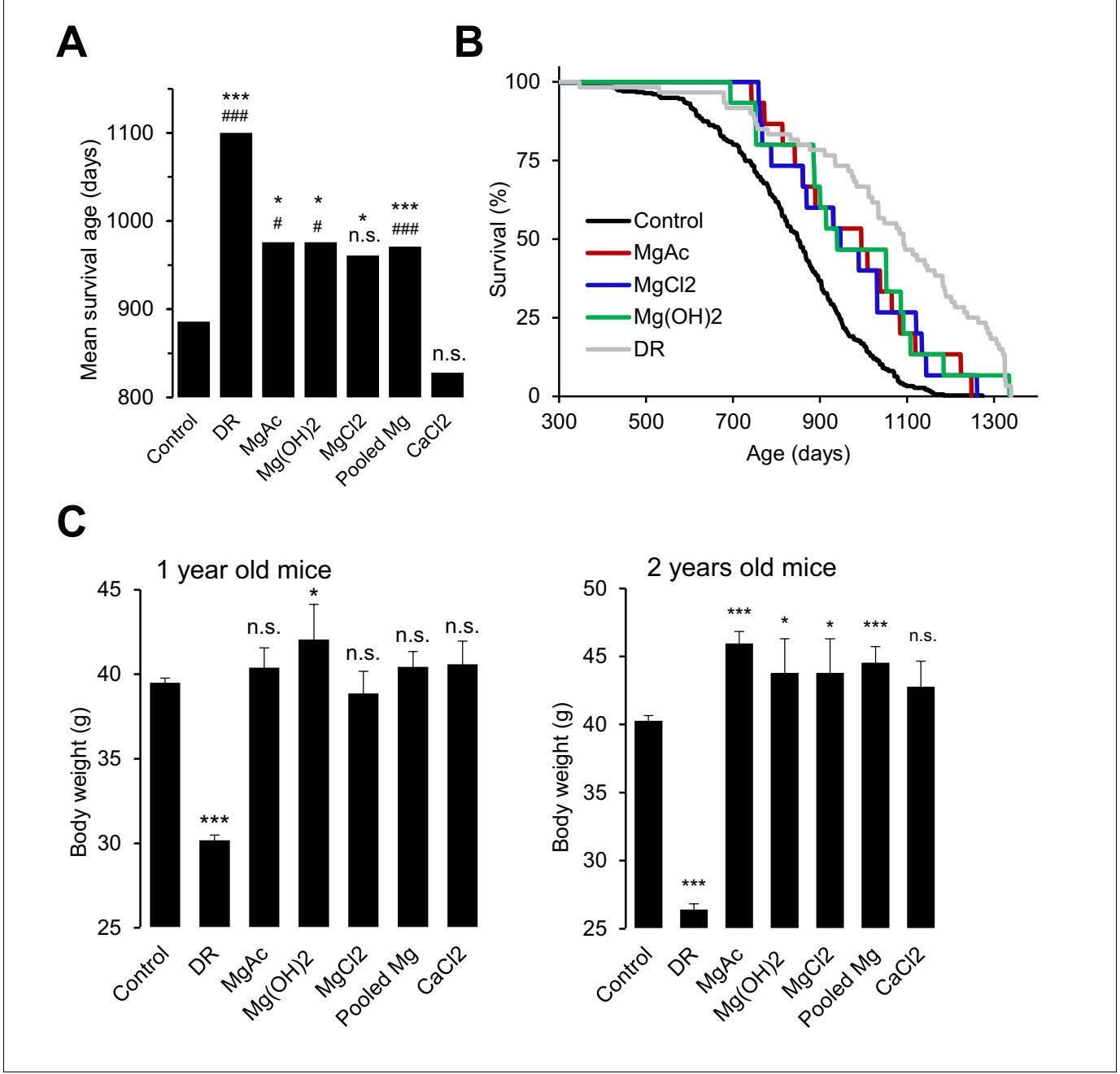

**Figure 7.** Effects of whole-life $Mg^{2+}$ dietary treatments on B6C3F1 mouse strain. (**A**) Mean survival ages of B6C3F1 mice maintained at a control diet (*Control*, n = 335), under dietary restriction (*DR*, n = 60), or supplemented by $Mg(CH_3COO)_2$ (*MgAc*, n = 15), $Mg(OH)_2$ (n = 15), $MgCl_2$ (n = 15) and $CaCl_2$ (n = 15) in drinking water as outlined in **Table 2**. *Pooled Mg* shows results for all $Mg^{2+}$ supplemented mice pooled within a common group (n = 45). The obtained survival distributions were analysed by the MATLAB computing environment to calculate mean lifespans and corresponding P-values: ***-$p \leq 0.001$; **-$p \leq 0.01$; *-$p \leq 0.05$; n.s. – not significantly different. Alternatively, survival data of control mice versus individually treated groups were assessed by log-rank test: ###-$p \leq 0.001$; ##-$p \leq 0.01$; #-$p \leq 0.05$; n.s. – not significantly different. (**B**) Kaplan-Meier survival distributions of B6C3F1 mice maintained on control diet (*Control*), $Mg^{2+}$ supplemented groups (*MgAc, MgCl_2, Mg(OH)_2*) or mice under dietary restriction (*DR*). (**C**) Body weights (mean+/-SEM) of control and nutritionally fortified mice studied in (**A**). ***-$p \leq 0.001$; **-$p \leq 0.01$; *-$p \leq 0.05$; n.s. – not significantly different (one-way ANOVA). n – number of mice examined.

**Table 2.** Dietary regimes used to maintain B6C3F1 mice.

| Experimental group | Chow 5 K54 | Drinking water (ad libitum) | Number of mice |
| --- | --- | --- | --- |
| Control | ad libitum | Regular water[*] | 335 |
| Dietary restriction | 60% of ad libitum | Regular water | 60 |
| Magnesium acetate supplemented | ad libitum | 1 g/l $Mg(CH_3COO)_2.(H_2O)_4$ in regular water | 15 |
| Magnesium chloride supplemented | ad libitum | 1 g/l $MgCl_2$ in regular water | 15 |
| Magnesium hydroxide supplemented | ad libitum | 14 mg/l $Mg(OH)_2$ in regular water[†] | 15 |
| Calcium chloride supplemented | ad libitum | 0.8 g/l $CaCl_2$ in regular water | 15 |

[*]Regular water contained 0.44 mg/l $Mg^{2+}$.

[†]$Mg(OH)_2$ concentration was lowered to prevent overall alkalization of the diet.

It has been reported that constitutive *Trpm6* inactivation leads to embryonic lethality, resulting in the generally accepted tenet that *Trpm6* is required for the development of the embryo proper (*Walder et al., 2009*). On the contrary, we provide genetic evidence that the embryonic mortality of *Trpm6*-deficient mice is caused by the loss of TRPM6 activity in placental SynT-I and yolk sac endoderm cells, and that *Trpm6*-deficient embryos are depleted of $Mg^{2+}$. Altogether, we postulate that *Trpm6* controls the maternal $Mg^{2+}$ supply to the fetus, and that growth failure and death are secondary phenotypes induced by $Mg^{2+}$ deprivation. Differences in the role of TRPM6 for embryonic survival of humans and mice may be attributable to different morphologies of the placental exchange interfaces, fetal growth rate, litter size and dietary preferences (*Simmons and Cross, 2005*). However, at present we cannot exclude that loss-of-function mutations in *TRPM6* might be associated with prenatal mortality in humans as well. Pioneering positional cloning studies (*Schlingmann et al., 2002*; *Walder et al., 2002*) and follow-up case reports (*Schlingmann et al., 2005*; *Konrad and Schlingmann, 2014*) focused on hospitalized human infants selected by the criterion of deleterious $Mg^{2+}$ deficiency. Hence, additional *TRPM6* phenotypes, such as infertility or embryonic death, might have been overlooked. Accordingly, it has recently been shown that single nucleotide polymorphisms in human *TRPM6* are associated with a neural tube closure defect, i.e. meningomyelocele (*Saraç et al., 2016*). Of note, dietary $Mg^{2+}$ supplementation is common practice of pregnant women (*Durlach et al., 2004*). Our work provides first mechanistic insight as to how this essential mineral is delivered to the fetus.

There is growing evidence to suggest that $Mg^{2+}$ deprivation is involved in the development of metabolic, immune, cardiovascular and neurological disorders (*de Baaij et al., 2015*). However, due to the lack of adequate mammalian genetic models, it is still unclear whether impaired $Mg^{2+}$ homeostasis can be regarded as the cause or the consequence of the latter pathophysiological processes. Therefore, we studied the impact of TRPM6 deletion on postnatal mice. *Trpm6*-deficient mice displayed shorter lifespans, failure to thrive, low physical activity, kyphosis, lung emphysema, sarcopenia and degeneration of lymphoid organs. In addition, *Trpm6*-deficient animals developed signs of catabolic metabolism such as lipodystrophy, increased insulin sensitivity and hypothermia, and showed suppression of the somatotropic axis accompanied by induction of xenobiotic detoxification gene networks in the liver. Altogether, we noted that the complex phenotype of *Trpm6*-deficient mice mirrors many phenotypic hallmarks of mutant mouse strains that are generally considered genetic models of 'accelerated aging' in the scientific literature (*López-Otín et al., 2013*). Furthermore, our unbiased screen for metabolic pathways dysregulated in *Trpm6*-deficient mice revealed an error in energy metabolism reminiscent of humans with mutations in the gene coding for mitochondrial carnitine palmitoyltransferase II, the most common inherited disorder of lipid metabolism in adult humans (*Bonnefont et al., 2004*; *Gempel et al., 2002*). Interestingly, an early biochemical study revealed that $Mg^{2+}$ and MgATP could regulate activity of mitochondrial carnitine acyltransferase activity (*Saggerson, 1982*). Accordingly, in proof-of-concept experiments, we demonstrated that $Mg^{2+}$ is specifically required for the utilization of acyl carnitines (AC) as an energy source in liver mitochondria. Hence, these results suggest that insufficient mitochondrial utilization of AC represents a plausible mechanism contributing to the pathologies developed by *Trpm6*-deficient mice. The exact role of $Mg^{2+}$ in AC metabolism as well as the molecular pathophysiology of carnitine

palmitoyltransferase II deficiency is still not completely understood and, being beyond of the scope of the current manuscript, has to be addressed in future studies.

Notably, if *Trpm6*-deficient mice were fed with a high $Mg^{2+}$ diet, they were viable and displayed normal physical activity, morphology of internal organs and tissue levels of AC, indicating that the phenotypes in *Trpm6*-deficient mice were triggered by $Mg^{2+}$ deficiency. Therefore, we studied in more detail how mutant mice develop organismal $Mg^{2+}$ deprivation. According to current thinking, the active transport of $Mg^{2+}$ in the kidney, in particular in DCT, determines the final urinary $Mg^{2+}$ concentration and is central to whole-body $Mg^{2+}$ balance (*de Baaij et al., 2015*). Contrary to this model, we demonstrate that global inactivation of *Trpm6* leads to $Mg^{2+}$ deficiency due to a defect in intestinal $Mg^{2+}$ uptake. To interrogate the renal role of TRPM6, we conditionally inactivated *Trpm6* in the kidney and, surprisingly, did not observe any changes in serum $Mg^{2+}$ levels. In contrast, intestine-specific disruption of *Trpm6* resulted in hypomagnesemia indicating that wildtype kidneys were not able to counterbalance the ablation of TRPM6 in the intestine. Taken together, our findings lend strong support to the new concept that *Trpm6*-dependent $Mg^{2+}$ uptake by the colon plays an indispensable role for systemic $Mg^{2+}$ balance in mice. However, we assume that renal TRPM6 activity may have an important role under conditions of insufficient dietary $Mg^{2+}$ intake, for instance, during prolonged fasting periods.

Currently there are two suggested mechanistic models of hypomagnesemia in humans carrying mutations in *TRPM6*. A pioneering study of patients with congenital hypomagnesemia anticipated that $Mg^{2+}$ malabsorption in the intestine plays a key role in organismal $Mg^{2+}$ deprivation (*Friedman et al., 1967*; *Milla et al., 1979*). More recently, it was reported that in affected humans renal $Mg^{2+}$ loss plays a key role (*Schlingmann et al., 2002*; *Walder et al., 2002*). The results obtained with *Trpm6*-deficient mice are well compatible with the former model. However, we cannot exclude physiological differences between humans and mice, an important issue to address in future studies. In conclusion, our findings support the idea that TRPM6 is a central gatekeeper of organismal $Mg^{2+}$ balance in mammals and that this role cannot be compensated for by any other $Mg^{2+}$ channel and transporter.

What is the molecular mechanism underlying insufficient $Mg^{2+}$ intake in *Trpm6*-deficient mice? TRPM6 and its close homolog TRPM7 have been invoked as molecular correlates of cation currents responsible for $Mg^{2+}$ entry into cells. In order to circumvent the limitations and pitfalls imposed by overexpression of recombinant proteins, we employed TS cells to compare the roles of native TRPM6 and TRPM7. We found that stem cells express both TRPM6 and TRPM7, thus mimicking the in vivo situation. Inactivation of *Trpm6* did not affect the self-renewal of TS cells. In contrast, *Trpm7*-deficient cells were not able to proliferate unless the cell culture medium was supplemented with additional $Mg^{2+}$, supporting the idea that TRPM7 plays a non-redundant role in cellular $Mg^{2+}$ uptake (*Schmitz et al., 2003*; *Chubanov et al., 2004*; *Ryazanova et al., 2010*). We further showed that wildtype TS cells exhibited $[Mg^{2+}]_i$- and $[MgATP]_i$-sensitive divalent cation currents, and that inactivation of *Trpm6* reduced current amplitudes. In contrast, deletion of TRPM7 caused complete ablation of these currents. Remarkably, ion currents in *Trpm6* deficient TS cells still expressing TRPM7 were substantially more sensitive to intracellular MgATP when compared to TRPM6/M7 co-expressing cells. Thus, in a native environment the presence of TRPM6 reduces the sensitivity of TRPM6/M7-like currents to inhibition by intracellular MgATP. It has recently been reported that the TRPM6/M7 heteromer is completely insensitive to MgATP after heterologous expression in HEK 293 cells (*Zhang et al., 2014*). Thus, native TRPM6/M7 currents do not fully recapitulate the latter findings obtained in a heterologous expression system and further studies are required to clarify these discrepancies. Nevertheless, our results are concordant with our model (*Chubanov et al., 2004*) that native TRPM6 functions primarily as a subunit of heteromeric TRPM6/M7 complexes by increasing current amplitudes and relieving TRPM7 from inhibition by $[MgATP]_i$ (*Figure 6G*). Such facilitated $Mg^{2+}$ entry is most probably not required for any cell autonomous function, but is necessary for epithelial $Mg^{2+}$ transport and the maintenance of serum $Mg^{2+}$ levels.

Inadequate nutritional $Mg^{2+}$ intake is commonplace in Western societies (in up to 68% of the US population [*King et al., 2005*]). In addition, a growing percentage of the population is exposed to drug-induced forms of hypomagnesemia (*de Baaij et al., 2015*). Consequently, we asked whether wildtype mice would benefit from extra $Mg^{2+}$ supplementation. Our proof-of-concept experiments suggest that life-long $Mg^{2+}$ supplementation may extend the lifespan of mice. This idea is concordant with the recent observation that $Mg^{2+}$ acting alone or in conjunction with dietary calorie

restriction counteracts lifespan-shortening effects of RNA-DNA hybrid (R loop) accumulation in yeasts and human HeLa cells (*Abraham et al., 2016*). Hence, large-scale investigations of nutritional $Mg^{2+}$ adjustments and their impact on health- and lifespan of different species should be enlightening in this regard.

## Materials and methods

### Mouse strains

Experiments involving animals were done in accordance with the EU Animal Welfare Act and were approved by the local councils on animal care (permit No 55.2-1-54-2532-134-13 from the Government of Oberbayern, Germany, and permit No 2347-15-2014 from the State Ministry of Brandenburg, Germany).

### *Trpm6* gene-trap mutant mouse strain

A mouse line carrying a gene-trap mutation in *Trpm6* (129S5/SvEvBrd-C57BL/6) (*Woudenberg-Vrenken et al., 2011*) was obtained from the Texas Institute for Genomic Medicine (stock No. TG0020; RRID:IMSR_TAC:tf0834). In the mutant allele (*Trpm6$^{\beta geo}$*), exons 2 and 3 of *Trpm6* were replaced by a *βgeo* reporter sequence (*βgeo*). The *βgeo* cassette includes a splice acceptor site sequence (SP), a bacterial *β*-galactosidase marker (*LacZ*), an internal ribosome entry site (*IRES*) and a neomycin resistance sequence (*NeoR*). Consequently, the *Trpm6$^{\beta neo}$* allele drives the expression of an aberrant transcript encoding only exon 1 of *Trpm6* spliced to *βgeo*. In order to obtain the *Trpm6$^{\beta geo}$* allele in a C57BL/6J genetic background, *Trpm6$^{\beta geo/+}$* 129S5/SvEvBrd-C57BL/6 mice were backcrossed with C57BL/6J mice for six generations. Mice were genotyped using PCR analysis of genomic DNA isolated from tail fragments. DNA was extracted and purified using the GenElute mammalian genomic DNA isolation kit (Sigma-Aldrich). DNA samples were examined by PCR using a set of allele-specific oligonucleotides (Metabion) and REDTaq DNA polymerase (Sigma-Aldrich). *Trpm6$^{\beta geo}$* allele was analyzed using primers Trpm6$^{\beta geo}$-Forward 5'-GCGTTGGCTACCCGTGAT-3' and Trpm6$^{\beta geo}$-Reverse 5'-CTGATAAGGAAGGCTGCTCTAAG-3' with PCR settings: 94°C 3', 94°C 30'', 55°C 30'', 72°C 1', 40 cycles, 72°C 5'. The amplified PCR product (367 bp) was visualised by standard *agarose* gel *electrophoresis and confirmed by sequencing.*

### Mice with a conditional *Trpm6* (*Trpm6$^{fl}$*) allele

A mouse line (C57BL/6) carrying a conditional mutation in *Trpm6* was generated by Taconic Artemis (Köln, Germany). Mouse genomic fragments corresponding to the targeted *Trpm6* segment were obtained from the C57BL/6J RPCIB-731 BAC library. The mutant locus contained two intronic LoxP sites flanking exon 17 of *Trpm6.* In addition, the targeting construct contained thymidine kinase (TK) and neomycin resistance (NeoR) markers allowing for negative and positive selection of mutant ES clones. NeoR was flanked by additional FRT recombination sites allowing for the subsequent deletion of NeoR sequences using Flp recombinase. BAC fragments, recombination sites, NeoR and TK were assembled in the targeting vector using an ET-cloning approach. The final targeting construct was confirmed by sequencing. C57BL/6N Tac embryonic stem (ES) cells were grown on mitotically inactive mouse embryonic fibroblasts (MEF) in DMEM (high glucose) containing 20% FBS (PAN) and 1200 µg/mL leukemia inhibitory factor (Millipore). The authentication of ES cells was performed by Taconic Artemis and no further authentication was performed by the authors. $10^7$ES cells were electroporated (Biorad Gene Pulser) with 30 µg linearized targeting vector. G418 (200 µg/mL) selection was started on day 2 and counter-selection with gancyclovir (2 µM) was started on day 5 after electroporation. ES clones obtained were examined for homologous recombination and single integration by Southern blotting (SB) using two probes located in 3' and 5' arms of the targeting construct and one additional probe located in NeoR. Sequences of the targeting vector, probes and SB images are available upon request. Mutant ES cells (clone 3339-AG-1) were injected into blastocysts isolated from uteri of 3.5 days post coitum (dpc) BALB/c females. The injected blastocysts were transferred to uteri of 2.5 dpc pseudopregnant females. Chimerism of the offspring obtained (G0) was evaluated by coat colour contribution (white/black). Highly chimeric G0 males were crossed with C57BL/6 females carrying a ubiquitously expressed *Flp* transgene to delete the NeoR cassette from

the mutant locus. Germline transmission in G1 offspring was identified by the presence of black coat colour followed by PCR analysis of genomic DNA isolated from tail fragments.

Deletion of exon 17 using Cre recombinase results in a frame-shift in the *Trpm6* transcript. The efficiency of Cre-mediated deletion of the floxed exon 17 (referred to as $Trpm6^{\Delta 17}$ allele) was examined using females ubiquitously expressing the *Sox2-Cre* transgene. The maternally inherited *Sox2-Cre* transgene allows for efficient recombination of LoxP-flanked genomic sequences in the single-cell embryo (*Hayashi et al., 2003*). This approach was employed to produce $Trpm6^{\Delta 17/+}$ individuals (*Table 1*). Mice were genotyped using PCR analysis of genomic DNA isolated from the tail fragments as described above for $Trpm6^{\beta geo/+}$ mice. $Trpm6^+$ allele was analyzed using primers $Trpm6^+$-Forward 5′-AGAGACGTGCAGTGTAGGACAGAG-3′ and $Trpm6^+$-Reverse 5′-ACGGCACACAGAAAA-CACCAG-3′ with PCR settings: 94℃ 3′, 94℃ 30′′, 64℃ 30′′, 72℃ 1′, 40 cycles, 72℃ 5′ (PCR product 549 bp). $Trpm6^{fl}$ allele was studied using primers $Trpm6^{fl}$-Forward 5′-GCAAATACAAGCAA-CACCTCC-3′ and $Trpm6^{fl}$-Reverse 5′-GAAGTTCCTATTCCGAAGTTCC-3′ with PCR settings: 94℃ 3′, 94℃ 30′′, 53℃ 30′′, 72℃ 1′, 40 cycles, 72℃ 5′ (PCR product 368 bp). $Trpm6^{\Delta 17}$ mutation was determined using primers $Trpm6^{\Delta 17}$-Forward 5′-TGTCTTCCATGTTGCTACGA-3′ and $Trpm6^{\Delta 17}$-Reverse 5′-CTTCCGGTCCACAGTTCAT-3′ with PCR settings: 94℃ 3′, 94℃ 30′′, 53℃ 30′′, 72℃ 1′, 40 cycles, 72℃ 5′ (PCR product 362 bp). Amplified PCR products were *confirmed by sequencing.*

## Mice with organ-restricted inactivation of *Trpm6*

Transgenic mice expressing Cre recombinase under the control of the mouse *Sox2* promoter (*Sox2-Cre* mice, C57BL/6J; RRID:IMSR_HAR:3359) (*Hayashi et al., 2003*), *Kidney-specific/Cadherin16* promoter (*Ksp-Cre* mice, C57BL/6J; RRID:MGI:4452131) (*Shao et al., 2002*) and *Villin1* promoter (*Villin1-Cre* mice, C57BL/6J; RRID:MGI:3581405) (*Madison et al., 2002*) were obtained from the Jackson Laboratory (Sacramento, CA, USA, stock No. 008454, 012237, and 004586 respectively). To conditionally inactivate *Trpm6*, *Sox2-Cre*, *Ksp-Cre* and *Villin1-Cre* mice were crossed with $Trpm6^{\Delta 17/+}$ mice to produce $Trpm6^{\Delta 17/+}$;*Sox2-Cre*, $Trpm6^{\Delta 17/+}$;*Ksp-Cre*, and $Trpm6^{\Delta 17/+}$;*Villin1-Cre* males. Next, these males were bred with $Trpm6^{fl/fl}$ females to generate offspring with a global, kidney and intestine-restricted disruption of *Trpm6* as outlined in *Table 1*. Inheritance of *Sox2-Cre*, *Ksp-Cre* and *Villin1-Cre* transgenes was determined by PCR analysis of tail DNA at conditions described previously (*Madison et al., 2002*; *Shao et al., 2002*; *Hayashi et al., 2003*).

## $Trpm7^{\Delta 17/+}$ mice

A mouse line carrying a conditional mutation in *Trpm7* ($Trpm7^{fl/fl}$ mice, 129 S6/SvEvTac; RRID:IMSR_JAX:018784) was kindly provided by David Clapham, Harvard Medical School, Boston, USA. The mutant locus contains two intronic LoxP sites flanking exon 17 of the *Trpm7* gene. Deletion of exon 17 results in a frame-shift mutation ($Trpm7^{\Delta 17}$ allele) and subsequent disruption of TRPM7. To produce $Trpm7^{\Delta 17/+}$ mice, $Trpm7^{fl/fl}$ females were crossed with *Sox2*-Cre males and the resulting offspring were genotyped by PCR as described previously (*Jin et al., 2008*). The generated $Trpm7^{\Delta 17/+}$ strain was maintained by intercross of $Trpm7^{\Delta 17/+}$ parents.

## Housing, metabolic and behaviour profiling of mice

### Housing conditions

Mice were kept in individually ventilated polycarbonate cages (IVC System, Tecniplast, Germany). Cages were changed weekly and were on a 12 hr light/dark cycle with artificial lighting. Temperature and relative humidity were 22 ± 1℃ and 50 ± 5%, respectively. Breeding animals were maintained on a multigrain chow Ssniff M-Z (Ssniff GmbH, Germany) and drinking water containing 20.1 mg/l $Mg^{2+}$ and 81.9 mg/l $Ca^{2+}$ (*ad libitum*). Litters were weaned at three weeks of age, genotyped and desired littermates were housed in cages as described above except that a 'maintenance' chow Sniff R/M-H (Ssniff GmbH, Germany) was used. Both Ssniff M-Z and Sniff R/M-H chows contained 0.22% $Mg^{2+}$ and 1.0% $Ca^{2+}$. In dietary $Mg^{2+}$ supplementation experiments, 4 week-old *Trpm6*-deficient and control mice were fed Ssniff R/M-H chow containing 0.75% $Mg^{2+}$ (Ssniff GmbH, Germany).

### Collection of specimen

Mice were weighed, killed and blood samples were collected by a cardiac puncture technique. Blood samples were incubated 30 min at RT, clots were removed by centrifugation (Heraeus Pico 17, 3500

rpm for 30 min at RT) and the resulting serum samples were stored at −80°C. Internal organs were removed, flash frozen in liquid nitrogen and stored at −80°C. Right tibias were dissected, cleaned from muscle tissues, dried overnight at 70°C and kept at RT. To determine urinary and fecal excretion rates of main minerals, mice were maintained for 24 hr in individual metabolic cages (Acme Metal Products, USA) under housing conditions as described above and supplied ad libitum with drinking water and chow. After 24 hr, the urine and feces produced were collected. Urine samples were stored at −80°C. Feces specimen were dried overnight at 70°C and stored at −20°C.

## Determination of $Mg^{2+}$ levels and other main elements
Content of main elements in bones, serum, urine, feces and gastrocnemius muscle samples was determined by inductively coupled plasma mass spectrometry (ICP-MS) by ALS Scandinavia (Sweden) as reported previously (*Rodushkin and Odman, 2001*; *Rodushkin et al., 2004*).

## Monitoring of food and water intake, locomotor activity, body lean mass, body temperature and energy content
Food and water intake were recorded with an automated Drinking and Feeding Monitor system (TSE Systems, Germany) with food baskets connected to weight sensors. An adaptation period of 2 days was followed by 3 days of data collection. Cumulative food intake and locomotor activity (based on infrared beams, InfraMot-Activity System, TSE Systems, Germany) were recorded. The core body temperature was measured using a rectal probe as reported previously (*Braun et al., 2009*). In vivo analysis of body lean mass was performed in conscious, restrained mice by nuclear magnetic resonance (EchoMRI[TM]-100H; EchoMRI LLC, USA) weekly and before metabolic measurements. The energy content in feces was determined by bomb calorimetry (IKA C5003; IKA Werke, Staufen, Germany).

## Evaluation of atherosclerosis development
Assessment of atherosclerosis lesions was performed as described previously (*van der Vorst et al., 2015*).

## Determination of insulin growth factor 1 (IGF1), major urinary proteins (MUPs), ß-hydroxybutyrate and glucose-tolerance test
Serum concentrations of IGF1 were analysed by a Mouse/Rat IGF-I ELISA kit (R and D-Systems). MUPs content as assessed by SDS-PAGE gel electrophoresis of 1 µl urine samples from individual mice. ß-Hydroxybutyrate was measured by a commercially available assay kit (Sigma-Aldrich). For a glucose-tolerance test, mice were fasted for 6 hr, followed by an oral gavage of glucose (2 mg/g body weight). Blood samples were collected via the tail vein and blood glucose levels were measured using a contour glucometer (Bayer, Germany). Plasma levels of insulin were measured using a commercially available ELISA kit (Alpco).

## Histological examination of tissues
### Hematoxylin-eosin staining of tissue sections
Tissues were fixed overnight at 4°C in 4% paraformaldehyde (PFA), dehydrated through a series of ethanol washes, cleared in three changes of xylene and embedded in paraffin. Tissue sections (5 µm) were cut by RM2125 RTS *microtome (Leica Microsystems,* Germany), mounted on Superfrost Plus slides (Menzel-Gläser), and dried at 70°C for 1 hr. Tissue sections were dewaxed in xylene, rehydrated through a series of ethanol washes to deionized water. Slides were incubated in Mayers hematoxylin solution (Carl Roth) for 10 min at RT followed by incubation in 0.5% Eosin Y solution (Carl Roth) for 13 min at RT. The colour reaction was stopped by deionized water, slides were dehydrated in ethanol, cleared in xylene and mounted using mounting medium (Carl Roth). Slides were examined using an Olympus CX41 microscope and Cell Imaging software (Olympus, Germany).

### Immunohistochemistry (IHC)
Polyclonal TRPM6-specific antibodies were raised by immunization of rabbits with H$_2$N-CERDK NRSSLEDHTRL-COOH peptide coupled via the N-terminus to keyhole limpet hemocyanin (KLH) and

purified by peptide affinity chromatography (Eurogentec, Belgium). Whole kidneys were dissected from eight week-old mice and embedded in Jung tissue freezing medium (Leica Microsystems, Germany). 10 µm cryosections were produced by a CM 3050S cryotom (Leica Microsystems, Germany), mounted on Superfrost Plus slides (Menzel-Gläser), air-dried for 20 min and fixed in 2% (w/v) para-formaldehyde in PBS (pH 7.4) for 20 min at RT. After washing in PBS (2×10 min), sections were blocked with 5% goat serum/0.5% Triton X100 (Sigma-Aldrich) in PBS for 2 hr at RT. The rabbit anti-TRPM6 antibody (1 µg/ml in 5% goat serum/0.5% Triton X100/PBS) was applied overnight at 4°C. Afterwards sections were washed in PBS (3×10 min, RT) and a goat anti-rabbit antibody conjugated with Alexa 488 (Life Technologies, Darmstadt, Germany; 1 µg/ml in 5% goat serum/PBS) was applied for 1 hr at RT. After washing in PBS (3×10 min, RT), sections were embedded in Dako Mounting Fluid (Dako Cytomation). Differential interference contrast and confocal images were obtained with a confocal laser scanning microscope LSM 540 META (Carl Zeiss, Germany). We used a Plan-Apochromat x63/1.4 oil objective, the 488 nm excitation wavelength of an argon laser, and a 505–570 nm band-pass filter. Acquired DIC and confocal images were analysed using the LSM 540 META software (Carl Zeiss, Germany).

## In situ hybridization (ISH)

cDNA templates for the production of cRNA *Trpm6* probes were produced by PCR with the following 2 sets of primers: Probe-1-Forward: 5'-aattaaccctcactaaagggGAGAGGAGGCCACAGTCAAG-3'; Probe-1-Reverse: 5'-taatacgactcactatagggGCTCAAAGACGATGTCACGA-3'; Probe-2-Forward: 5'-aattaaccctcactaaagggCCTGTCAAAGAAGAAGAGGAA-3'; Probe-2-Reverse: 5'-taatacgactcactataggg-gAGAAAAGACTTCACAATG-3'. Primers contained T7 (reverse primer lower case) or T3 (forward primer lower case) RNA polymerase sites. PCR products were gel purified (Qiagen Gel Extraction Kit) and sequence verified (ARGF). Digoxigenin (DIG) labelled cRNA probes were synthesized according the manufacturer's instructions (Roche, 10x DIG RNA labelling kit). Both probe sets produced similar results on n = 5 placentas at each gestational time point examined, and n = 3 WT kidney or intestinal samples. *Gcm1* and *SynA* riboprobes were described previously (*Dawson et al., 2012*). Preparation of tissue sections and ISH procedures were performed as previously described (*Dawson et al., 2012*). Slides were imaged by an Aperio slide scanner and analysed using Image-Scope software.

## Whole-genome profiling of the liver transcriptome in *Trpm6*-deficient mice

Microarray data were deposited in NCBI Gene Expression Omnibus (GEO) (GSE70457). Liver tissues were collected from 12–13 week-old *Trpm6*-deficient (*Trpm6^{Δ17/Δ17};Sox2-Cre*, n = 3) and control (*Trpm6^{+/fl}*, n = 4) male littermates, snap-frozen in liquid nitrogen and stored at −80°C. Total RNA was extracted using the GenElute mammalian total RNA purification kit (Sigma-Aldrich). Whole genome profiling was performed using a GeneChip Mouse Gene 1.0 ST Array (Affymetrix) at Source Bioscience (Berlin, Germany). Biotinylated single-stranded DNA was prepared according to the standard Affymetrix protocol (Whole Transcript Expression arrays) from 100 ng total RNA using the WT terminal labelling kit. 2.5 µg of fragmented and labelled ssDNA were hybridized for 16–18 hr at 45°C. GeneChips were washed and stained in an Affymetrix Fluidics Station 450. GeneChips were scanned using the Affymetrix GeneChip Scanner 3000. Processing of the array data, including quality assessment, background correction, normalization and summarization was performed with the Affymetrix Expression Console (version 1.4.0). All statistical analyses were carried out with the statistical computing environment R (version 3.1.2, www.R-project.org). Differential expression analysis was performed with the R package limma (version 3.22.4) (*Ritchie et al., 2015*). p-values were adjusted for multiple testing with the Benjamini-Hochberg method for controlling the false discovery rate (FDR). A heatmap was generated for a group of 46 transcripts differentially expressed at a level of FDR $p \leq 0.1$. Analysis of the affected pathways and causal transcriptional regulators was performed by Ingenuity pathway analysis (IPA) environment (www.ingenuity.com, RRID:SCR_008653) using a set of 2443 transcripts changed at $p \leq 0.05$ confidence level (t-test).

## Metabolomic profiling of serum, liver and skeletal muscle in *Trpm6*-deficient mice

Serum, liver and gastrocnemius muscle samples were collected from 8–10 week-old control (*Trpm6*$^{+/}$$^{fl}$, n = 8) and *Trpm6*-deficient (*Trpm6*$^{Δ17/Δ17}$;*Sox2-Cre*, n = 6) male littermates, flash frozen in liquid nitrogen and stored at −80°C. Metabolomic analyses were performed at Biocrates Life Sciences AG (Innsbruck, Austria). Measurements comprised the quantification of 237 metabolites including 41 amino acids/biogenic amines, 40 acylcarnitines (AC), 22 bile acids (BA), 14 lysophosphatidylcholines (LysPC), 77 phosphatidylcholines (PC), 15 sphingomyelins (SM), 17 eicosanoids/prostaglandins and 11 energy metabolism intermediates as outlined in *Supplementary file 3*. To extract metabolites, tissue samples were treated with corresponding extraction buffers and incubated in a chilled ultra-sonic bath for 5 min. Afterwards samples were centrifuged and the supernatant was used for analysis. FIA- and LC-MS/MS measurement techniques were applied as described in detail previously (*Pena et al., 2014*). All statistical analyses have been applied by using the statistical computing environment R (www.r-project.org). Metabolite measurements containing more than 75% missing values or more than 75% of values below the limit of detection (LOD) across all samples per matrix were removed from analysis. Measured concentrations of metabolites were log$_2$-transformed for moderated statistical tests. Measurements were scaled to μ = 0 mean and unit standard deviation for each biological matrix separately for heatmap visualization. Changes in average metabolite levels between control and mutant individuals were tested using a linear model framework implemented in the R package Limma (*Smyth, 2004*). Resulting P-values for moderated t-test were corrected for multiple testing by Benjamini and Hochberg approach. A p-value threshold of 0.05 was considered as significant. A Heatmap diagram for metabolites with significant changes was calculated with the R-package Heatmap using ward clustering and Euclidean distance measure and the R-package Venn-nDiagram was applied to calculate a Venn diagram of significantly changed metabolites across sample matrices.

## Mitochondrial isolations and analyses

Mouse liver mitochondria were isolated by differential centrifugation from freshly prepared homogenates as previously described (*Springer et al., 2002*; *Schulz et al., 2015*). Liver mitochondria were further purified by Percoll density gradient centrifugation (*Schulz et al., 2013*). Isolated mitochondria were subjected to quantification by the Bradford assay and kept on ice until use. Assessment of the mitochondrial membrane potential $Δψ_m$ (MMP) was followed by Rh123 fluorescence quenching (Ex. 485 nm, Em. 528 nm) in a 96-well plate reader (BioTek) and quantitatively evaluated by curve analysis as previously described (*Schulz et al., 2013*) (set threshold slopes were $\geq$0.67 for start points and $\leq$0.67 for end points, respectively). In order to exclude fluctuations at measurement start, slope calculations were started after 6 min measurement time with slope values $\leq$ 1.5. A kit-based assay (ATP Bioluminescence Assay Kit, Roche) was used to analyze the ATP content from cleared lysates after 30 min mitochondrial ATP synthesis at RT, initiated by the addition of 160 μM ADP and stopped at 95°C for 5 min. For both analyses assay buffer composition was 0.2 M sucrose, 10 mM MOPS-Tris, 1 mM Pi and 10 μM EGTA. Respiratory substrates were either succinate (25 mM)/rotenone (2 μM), or DL-octanoylcarnitine (10 μM)/malate (12.5 mM), or DL-palmitoylcarnitine (10 μM)/malate (12.5 mM). Buffers and solutions were essentially $Mg^{2+}$-free, as determined by ICP-OES (Ciros Vision, SPECTRO Analytical Instruments GmbH) after wet ashing with 65% nitric acid (*Zischka et al., 2011*). EDTA, $Mg^{2+}$, $Ca^{2+}$, or $Zn^{2+}$ was added at the concentrations indicated in the respective figures.

## Isolation and characterization of mouse trophoblast stem (TS) cells

### Isolation of TS cells

TS cells were isolated as described (*Tanaka et al., 1998*), with several modifications (*Erlebacher et al., 2004*; *Natale et al., 2009*). 3.5 *days post-coitum* blastocysts were isolated from *Trpm6*$^{βgeo/+}$ parents. Individual blastocysts were incubated in a humidified cell culture incubator (Heraeus, Thermo Fisher Scientific) at 37% and 5% $CO_2$ for 3 days in 12 well plates (Sarstedt) containing 8×10$^4$/well irradiated mouse embryonic fibroblasts (MEFs) (Millipore) in RPMI 1640 medium (Life Technologies) supplemented with 20% fetal bovine serum (ES type, Life Technologies), 1 mM sodium pyruvate (cell culture type, Sigma-Aldrich), 100 μM $β$-mercaptoethanol (Sigma-Aldrich), 50

μg/ml streptomycin and 50 U/ml penicillin (all from Life Technologies), 1.0 μg/ml heparin (cell culture type, Sigma-Aldrich), 25 ng/ml human recombinant FGF4 (R and D systems), 5 ng/ml human recombinant TGF-$\beta$1 (R and D systems), 10 ng/ml recombinant activin A (R and D systems) and an additional 10 mM MgCl$_2$. The attached embryos were disaggregated by 0.05% trypsin-EDTA (Life Technologies) and derived cells were further co-cultured with MEFs as described above. The obtained TS cells were propagated and adapted to MEF-free conditions without additional Mg$^{2+}$ in the culture medium. The authentication of TS cells was based on expression of a trophoblast stem cell marker *Esrrb*, characteristic morphological appearance and ability to proliferate only in the presence of FGF4 and TGF-$\beta$1 (*Tanaka et al., 1998*; *Simmons and Cross, 2005*). TS cells were tested negative for mycoplasma contamination using QuickTest kit (Biotool).

Genotypes of TS cells were determined by PCR analysis of genomic DNA using conditions described above for *Trpm6$^{\beta geo/+}$* mice. TS cells were further examined by RT-PCR. Total RNA was extracted from TS cell pellets using the GenElute mammalian total RNA purification kit (Sigma-Aldrich). First strand cDNA synthesis was performed by RevertAid H minus reverse transcriptase (Thermo Scientific). PCR was performed using REDTaq DNA polymerase (Sigma-Aldrich) with two primer sets: Trpm6a-Forward 5′-GCTGCCAAATCTGCCACAAT-3′ and Trpm6a-Reverse 5′-TGCCCA-CAGTCCCATCATCACA-3′ or Trpm6b-Forward 5′-CCAGCTCAAAAGACCCTCACAGATGC-3′ and Trpm6b-Reverse 5′-CACACCACATCTTTTCCGACCAG-3′. The following PCR conditions were used: 94°C 3′, 94°C 30′′, 56°C 30′′, 72°C 1′, 35 cycles, 72°C 5′. Amplified PCR products were 651 bp or 586 bp, respectively. Self-renewal of *Trpm6$^{\beta geo/\beta geo}$* TS cells was assessed by determination of DNA content as described previously (*Tanaka et al., 1998*). Briefly, TS cells were cultured for 3 days, dissociated by trypsin-EDTA and fixed with 40% ice-cold ethanol. Fixed TS cells were incubated in propidium iodide (PI) staining solution (50 μg/ml PI, 0.2 mg/ml RNaseA in PBS, all from Sigma-Aldrich) for 30 min at RT. Stained cells were dissolved in PBS and examined using BD FACSCalibur (BD Biosciences, Germany) and FlowJo software (www.flowjo.com).

*Trpm7*-deficient and corresponding control TS cells were derived as described above using *Trpm7$^{\Delta 17/+}$* mice. The obtained *Trpm7$^{+/+}$* and *Trpm7$^{\Delta 17/\Delta 17}$* TS cells were propagated and adapted to MEF-free conditions in cell culture medium supplemented with 10 mM MgCl$_2$. Genotypes of TS cells were determined by PCR analysis of genomic DNA using conditions described for *Trpm7$^{\Delta 17/+}$* mice. The lack of *Trpm7* transcripts in *Trpm7$^{\Delta 17/\Delta 17}$* TS cells was verified by RT-PCR using primers Trpm7-Forward 5′-AGTAATTCAACCTGCCTCAA-3′ and Trpm7-Reverse 5′-ATGGGTATCTCTTCTG TTATGTT-3′ and the following PCR conditions: 94°C 3′, 94°C 30′′, 50°C 30′′, 72°C 1′, 35 cycles, 72°C 5′. The amplified PCR product was 287 bp.

To study growth rates, TS cells of each genotype were seeded in 6-well plates ($1\times10^5$ cells/well) in cell culture medium containing 10 mM MgCl$_2$. After 24 hr (day 1), the culture medium was replaced with fresh medium either with or without 10 mM MgCl$_2$ and the cells were further cultured for additional 3 days. The cell density was determined at 24 hr intervals using a Neubauer chamber (Marienfeld Superior). To calculate growth rates, the cell number at day 1 was designated as 100%. The experiment was repeated three times and a Student's *t*-test was applied to compare the growth rates of control versus mutant cells.

## Electrophysiology

Whole-cell currents were measured using an EPC10 patch-clamp amplifier and PatchMaster software (Harvard Bioscience, Germany). Voltages were corrected for a liquid junction potential of 10 mV. Currents were elicited by a ramp protocol from −100 mV to +100 mV over 50 ms acquired at 0.5 Hz and a holding potential of 0 mV. Inward and outward current amplitudes were extracted at −80 mV and +80 mV and were normalized to cell size as pA/pF. Capacitance was measured using the automated capacitance cancellation function of EPC10. Patch pipettes were made of borosilicate glass (Science Products) and had resistances of 2–3.5 MΩ. The standard extracellular solution contained (in mM): 140 NaCl, 3 CaCl$_2$, 2.8 KCl, 10 HEPES, and 11 glucose (all from Sigma-Aldrich). A divalent-free (DVF) extracellular solution contained (in mM): 140 NaCl, 2.8 KCl, 10 HEPES, 11 glucose and 5 EDTA. TRPM6/M7-like currents were induced by Mg$^{2+}$-free intracellular solution, containing (in mM): 140 Cs-glutamate, 8 NaCl, 10 EGTA, 5 EDTA and 10 HEPES. All solutions were adjusted to pH 7.2 using a FE20 pH-meter (Mettler Toledo, Germany). The osmolality of all solutions was adjusted to

290 mOsm using Vapro 5520 osmometer (Wescor Inc., USA). Data were compared by an unpaired Student's $t$-test.

For $Mg^{2+}$ ($[Mg^{2+}]_i$) and MgATP ($[MgATP]_i$) dose responses, the intracellular pipette solution contained (in mM): 120 Cs-glutamate, 8 NaCl, 10 HEPES, 2.7 EDTA and various amounts of $MgCl_2$ or MgATP (Sigma-Aldrich). The solutions were adjusted to pH 7.2 and 290 mOsm. Concentrations of MgATP and free $Mg^{2+}$ were calculated using WebMaxC (maxchelator.stanford.edu). To determine $IC_{50}$ values, datasets were fitted using a nonlinear (least-squares) regression analysis (GraphPad Prism 6.0 software) and the following equation:

$$E(c) = E_{min} + (E_{max} - E_{min})/(1 + 10^{\wedge}((IC_{50} - C)h))$$

with E being the effect/current at a given concentration C of inhibitor, $E_{min}$ the minimal effect/current, $E_{max}$ the maximally achievable effect, $IC_{50}$ the half-maximal concentration and h the Hill slope factor. Statistical analysis of dose-response curves and $IC_{50}$ values was performed using the extra sum-of-squares F test with the threshold P value 0.05 (GraphPad Prism 6.0).

## Characterization of *TRPM7*-deficient human haploid leukaemia (HAP1) cells

### Isolation and maintenance of *TRPM7*-deficient *HAP1* cells

Wildtype parental cells (clone C631) and *TRPM7*-deficient (clone 10940–04) HAP1 cells were acquired from Horizon Genomics (Vienna, Austria). The authentication of HAP1 cells was performed by Horizon Genomics (*Essletzbichler et al., 2014*) and no further authentication was performed by the authors. A CRISPR/Cas9 approach was used to introduce a 17 bp (GTGACCATTTTAATCAG) deletion in exon 4 of the human *TRPM7* gene resulting in a frame-shift mutation. Genotypes of HAP1 cells were confirmed by PCR amplification of genomic DNA using primers hTRPM7-Forward 5'-TATTTGTATGCACCTTTGTA-3' and hTRPM7-Reverse 5'-TGTTTTAATCTCACCTTTTT-3' with PCR parameters: 94°C 3', 94°C 30'', 50°C 30'', 72°C 1', 40 cycles, 72°C 5'. PCR products (364 bp and 347 bp in wild type and mutant clones, respectively) were confirmed by sequencing. HAP1 cells were tested negative for mycoplasma contamination using QuickTest kit (Biotool).

HAP1 cells were cultured in Iscove's Modified Dulbecco's Medium (IMDM) supplemented with 10% FBS and 100 U/ml penicillin, 100 µg/ml streptomycin and 10 mM $MgCl_2$ (all from Thermo Fisher Scientific). Cells were maintained in a humidified cell culture incubator (Heraeus, Thermo Fisher Scientific) at 37°C and 5% $CO_2$. Western blot analysis was performed as described previously (*Nörenberg et al., 2016*). Examinations of growth rates and endogenous TRPM7-like currents were conducted as described above for TS cells.

### Determination of total $Mg^{2+}$ content

HAP1 cells of each genotype were grown in T175 $cm^2$ flasks (Sarstedt) in 10 mM $MgCl_2$ supplemented cell culture medium as described above. At ~50% confluence the medium was replaced with fresh medium without 10 mM $MgCl_2$ and the cells were cultured for additional 24 hr. Next, cells were washed with PBS, disaggregated by trypsinization and collected in 50 ml plastic tubes (Sarstedt). After centrifugation (3 min, 1000 rpm), the cell pellet was resuspended in 1 ml PBS and passed to a fresh 1.5 ml tube. The cell suspension was centrifuged (3 min, 3500 rpm), supernatant was completely removed and the cell pellet was dried overnight at 70°C. The dried cell pellet was analysed by ICP-MS in ALS Scandinavia (Sweden). The experiment was repeated four times.

### Assessment of ATP levels

HAP1 cells of each genotype were seeded in 96-well plates (white wall/clear bottom, type 3610, Costar) at a density of $5 \times 10^5$ cells/well in the 10 mM $MgCl_2$ supplemented cell culture medium (100 µl/well); eight wells per genotype were used. After 24 hr, the cell culture medium was replaced with fresh regular medium (without additional 10 mM $Mg^{2+}$) and the cells were cultured for additional 24 hr. To determine cell viability, 10 µl of CCK-8 reagent (Cell Counting Kit-8) was added in four wells per genotype. The plates were incubated for 3 hr in the cell culture incubator (Heraeus, Thermo Fisher Scientific) at 37°C and 5% $CO_2$. Next, 96-well plates were incubated at room temperature (RT) for 30 min and 100 µl of CellTiter-Glo2.0 reagent (Promega) was added to the remaining four wells per genotype. After 10 min incubation at RT, either ATP-induced luminescence or CCK-8

absorbance (450 nm) were determined using a plate reader (FLUOstar Omega, BMG Labtech). CCK-8 absorbance was used to normalize ATP-induced bioluminescence. The normalized bioluminescence of wildtype cells was designated 100%. The experiment was repeated six times.

## High-resolution respirometry of living HAP1 cells

Oxygen consumption of HAP1 cells was assessed by Oxygraph-2k measurements (Oroboros Instruments GmbH, Austria) as described previously (Pesta and Gnaiger, 2012). Briefly, WT and KO HAP1 cells were maintained in standard medium supplemented by 10 mM $MgCl_2$. The $Mg^{2+}$ supplemented medium was replaced with regular medium (without additional 10 mM $Mg^{2+}$) and the cells were cultivated for a further 24 hr. Next, oxygen flux from routine respiration of $1.5 \times 10^6$ cells at 37°C and maximal oxygen flux after stepwise CCCP addition (2 µl steps from 1 mM stock solution) were determined. The oxygen flux was baseline-corrected for non-mitochondrial oxygen consuming processes by the addition of 0.5 µM rotenone (complex I inhibitor, Sigma-Aldrich) and 2.5 µM antimycin A (complex III inhibitor, Sigma-Aldrich).

## Whole-life dietary treatments of B6C3F1 mice

Mice were raised in a specific pathogen-free facility at Jackson Laboratory Sacramento (Sacramento, CA, USA). The long-lived B6C3F1 hybrid strain (Lipman et al., 1999; Turturro et al., 1999) was used. Specifically, F1 females were derived by crossing C57BL/6J females with C3H/HeJ males. The produced B6C3F1 females were weaned at 3 weeks of age, and afterwards kept in individually ventilated polycarbonate cages (Thoren Caging Systems) on multigrain chow 5 K54 (Purina) containing 0.22% $Mg^{2+}$ and drinking water containing 0.44 mg/l $Mg^{2+}$. Five mice were housed per cage and cages were changed every two weeks. The housing rooms were with artificial lighting and 12 hr light/dark cycle (6 am to 6 pm). Temperature and relative humidity in animal rooms were 22 ± 4°C and 50 ± 15%, respectively. At 5 months of age, mice were assigned to control, dietary restricted (DR) or supplemented cohorts as outlined in Table 2. For supplementation experiments, corresponding salts (Sigma-Aldrich) were diluted in drinking water that was administered ad libitum throughout the lifespan of all groups (Table 2). To reduce bias, the supplemented and control groups were examined simultaneously and the study was performed as a blinded trial. Mice were inspected daily. Necropsies of randomly selected dead mice revealed that 58 of 88 control females (66%) developed tumours. 8 of 11 females (73%) from the three $Mg^{2+}$ supplemented groups also had tumours suggesting that high dietary $Mg^{2+}$ had no substantial effect on tumor rate at death. Kaplan-Meier survival distributions were computed to illustrate survival times. For statistical analysis, we used MATLAB computing environment and programming language (MathWorks, www.de.mathworks.com), in particular its built-in fast convolution function that allows to find the distribution of the sum of independent random variables, given the distributions of individual variables. To enable statistical comparisons between control and dosed groups, the distribution of the mean lifespan of 15 control mice was found as a convolution of the original distribution of lifespans of all control mice. To calculate P-values of the mean lifespan from the dosed groups relative to the control group, we calculated the probability of the average of 15 control mice showing the same or more extreme (away from control mean) lifespan than the experimentally determined mean lifespan of the dosed mice. A similar technique was used to find P-values for pooled Mg and dietary restriction groups relative to controls. The MATLAB code is available from the Dryad Digital Repository (Chubanov and Gudermann, 2016). In addition, the survival data of control mice versus individual treated groups were assessed by log-rank test using GraphPad Prism software. P-values for both approaches are indicated in Figure 7A.

## Acknowledgements

We thank Jackson Laboratory Sacramento for the help with whole-life dietary treatments of mice. We thank David Clapham for providing $Trpm7^{fl/fl}$ mice; Ilia Rodushkin for the support in ICP-MS; Marc Freichel and Petra Weißgerber for their help to isolate TS cells; Fabian Bamberg and Mike Notohamiprodjo for the help with X-ray imaging of mice; Sabine Schmitt for the help with high-resolution respirometry. We thank Renate Heilmair and Joanna Zaißerer for technical assistance. VC, SZ and TG were supported by the Deutsche Forschungsgemeinschaft, TRR 152. WJ and AS were supported by the German Federal Ministry of Education and Research (BMBF, DZD). SZ was supported

by Marie-Curie Fellowship (REA) FP7-PEOPLE-2012-CIG. EPCV and WC were supported by the Deutsche Forschungsgemeinschaft (SFB1123-A1) and German Centre for Cardiovascular Research (MHA VD1.2).

## Additional information

### Funding

| Funder | Grant reference number | Author |
|---|---|---|
| Deutsche Forschungsgemeinschaft | TRR 152-P15 | Vladimir Chubanov Thomas Gudermann |
| Deutsche Forschungsgemeinschaft | SFB1123-A1 | Emiel PC van der Vorst Christian Weber |
| Deutsche Forschungsgemeinschaft | TRP 152-P14 | Susanna Zierler |
| Seventh Framework Programme | Marie-Curie Fellowship FP7-PEOPLE-2012-CIG | Susanna Zierler |

The funders had no role in study design, data collection and interpretation, or the decision to submit the work for publication.

### Author contributions

VC, TG, Conception and design, Acquisition of data, Analysis and interpretation of data, Drafting or revising the article, Contributed unpublished essential data or reagents; SF, AW, CL, CE, WJ, HB, AB, BA, LM, LS, EPCvdV, Acquisition of data; DGS, AGR, Conception and design, Acquisition of data, Analysis and interpretation of data, Drafting or revising the article; YS, Acquisition of data; Analysis and interpretation of data; FT, VJ, CW, Analysis and interpretation of data; ÖAY, KS, Acquisition of data, Analysis and interpretation of data; AS, HZ, Conception and design, Acquisition of data, Analysis and interpretation of data; SZ, Acquisition of data, Analysis and interpretation of data, Drafting or revising the article

### Author ORCIDs

Vladimir Chubanov, http://orcid.org/0000-0002-6042-4193
David G Simmons, http://orcid.org/0000-0002-4115-9371
Susanna Zierler, http://orcid.org/0000-0002-4684-0385

### Ethics

Animal experimentation: Experiments involving animals were done in accordance with the EU Animal Welfare Act and were approved by the local councils on animal care (permit No 55.2-1-54-2532-134-13 from Government of Oberbayern, Germany, and permit No 2347-15-2014 from State Ministry of Brandenburg, Germany).

## Additional files

### Supplementary files

• Supplementary file 1. Whole genome profiling of hepatic transcripts altered in *Trpm6*-deficient mice. Dataset is available in the Dryad Digital Repository (*Chubanov and Gudermann, 2016*). (1) Genome-wide analysis of hepatic transcriptome in control vs *Trpm6*-deficient mice. (2) Up- and down-regulated transcripts in the liver of *Trpm6*-deficient mice with the false discovery rate (FDR) $p \leq 0.1$.

• Supplementary file 2. Ingenuity Pathway Analysis (IPA) analysis of hepatic transcripts altered in *Trpm6*-deficient mice. Dataset is available from the Dryad Digital Repository (*Chubanov and Gudermann, 2016*). (1) IPA Canonical Pathways representing differentially expressed genes in the

liver of *Trpm6*-deficient mice. (2) IPA Causal Networks for differentially expressed genes in the liver of *Trpm6*-deficient mice.

• Supplementary file 3. Metabolic profiling of the serum, liver and gastrocnemius muscle of *Trpm6*-deficient mice. Dataset is available in the Dryad Digital Repository (*Chubanov and Gudermann, 2016*). (1) Statistical analysis of metabolite measurements in serum samples. (2) Statistical analysis of metabolite measurements in gastrocnemius muscle samples. (3) Statistical analysis of metabolite measurements in liver samples. (4) Metabolites significantly changed in serum samples of *Trpm6*-deficient mice (FDR p≤0.05). (5) Metabolites significantly changed in gastrocnemius muscle samples of *Trpm6*-deficient mice (FDR p≤0.05). (6) Metabolites significantly changed in liver samples of *Trpm6*-deficient mice (FDR p≤0.05). (7) Abbreviations of metabolites.

## Major datasets

The following datasets were generated:

| Author(s) | Year | Dataset title | Dataset URL | Database, license, and accessibility information |
|---|---|---|---|---|
| Chubanov V, Gudermann T | 2015 | Whole-genome profiling of the liver transcriptome in Trpm6 gene deficient mice and control littermates | https://www.ncbi.nlm.nih.gov/geo/query/acc.cgi?acc=GSE70457 | Publicly available at the NCBI Gene Expression Omnibus (accession no: GSE70457) |
| Chubanov V, Gudermann T | 2016 | Data from: Epithelial magnesium transport by TRPM6 is essential for prenatal development and adult survival | http://dx.doi.org/10.5061/dryad.gs7fv | Available at Dryad Digital Repository under a CC0 Public Domain Dedication |

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
