## [Decision Letter]

Thank you for submitting your article "Epithelial magnesium transport by TRPM6 is essential for prenatal development and adult survival" for consideration by *eLife*. Your article has been favorably evaluated by Richard Aldrich (Senior Editor) and three reviewers, one of whom, Richard S Lewis (Reviewer #1), is a member of our Board of Reviewing Editors. The following individual involved in review of your submission has agreed to reveal their identity: Thomas Voets (Reviewer #2).

The reviewers have discussed the reviews with one another and the Reviewing Editor has drafted this decision to help you prepare a revised submission.

Summary:

In this paper, Chubanov et al. provide an extensive analysis of the consequences of TRPM6 deficiency in mice, from embryonic development through adulthood. TRPM6 is considered an important regulator of magnesium transport, based on the severe hypomagnesemia observed in patients lacking functional TRPM6 and on Mg-permeable and -regulated ion channel activity in cells heterologously expressing TRPM6. However, the physiological mechanisms and functions of TRPM6 have been difficult to determine because global deletion of TRPM6 in mice causes embryonic death. The authors have used targeted TRPM6 deletions in mice to reveal that embryonic lethality is caused by TRPM6 deficiency in extraembryonic tissues rather than the embryo itself. By targeting the TRPM6 knockout to the embryo, they create a useful model system with which to study the phenotype of TRPM6 postnatally and in adult mice. A detailed characterization of these mice shows that TRPM6-dependent Mg^2+^ uptake in the colon but not kidney is essential for maintaining systemic Mg^2+^ homeostasis, and hypomagnesemia in TRPM6-deficient mice impairs growth, metabolism and lifespan, attributable to the impaired utilization of long-chain fatty acids by mitochondria to make ATP. Finally, dietary Mg^2+^ supplements, in addition to reversing the multiple effects of TRPM6 deletion, also prolong the lifespan of wild type mice, an intriguing observation that awaits replication in other organisms.

All three reviewers agree that the paper is extensive and significantly extends the understanding of TRPM6's role in Mg^2+^ homeostasis pre- and postnatally, and that the data are generally of high quality and clearly presented. No major new experiments are required, but some straightforward measurements described below would provide direct evidence needed to strengthen two important mechanistic conclusions.

Essential revisions:

1) The authors suggest that Mg^2+^ deprivation in TRPM6- deficient mice impairs ATP levels by preventing the use of acylcarnitine as a substrate. The link between Mg^2+^, AC metabolism and ATP is only shown for isolated mitochondria and nonphysiological concentrations of [Mg^2+^]. A simple measurement of ATP or mitochondrial potential (using fluorescent dyes) in intact cells from WT vs. KO mice would provide direct support to strengthen this conclusion.

2) The authors propose that TRPM6 regulates Mg^2+^ transport by reducing the MgATP-dependent inhibition of heteromeric TRPM6/7 channels. First, the evidence for this rests on whether the inhibition curves in Figure 6 are significantly different between WT and KO animals; this should be evaluated. In addition, it is not clear that MgATP in intact cells is in the required range that would differentially affect the current in the WT vs. TRPM6 groups (Figure 6); either the ATP level should be measured, or a suitable reference provided. Finally, another group (Zhang, Yu et al., 2014) reported that TRPM6 expression by itself produced currents, and TRPM6 and M7 coexpression generated currents that were essentially insensitive to MgATP inhibition. This should be discussed in the context of the results shown in Figure 6, where TRPM7 KO completely eliminated current, and wild type and TRPM6 currents are both inhibited by MgATP.

---

## [Author Response]

*Essential revisions:*

*1) The authors suggest that Mg^2+^ deprivation in TRPM6-deficient mice impairs ATP levels by preventing the use of acylcarnitine as a substrate. The link between Mg^2+^, AC metabolism and ATP is only shown for isolated mitochondria and nonphysiological concentrations of [Mg^2+^]. A simple measurement of ATP or mitochondrial potential (using fluorescent dyes) in intact cells from WT vs. KO mice would provide direct support to strengthen this conclusion.*

Thank you for this important point. The studies with isolated liver mitochondria were conducted because our metabolic profiling experiments suggested that mitochondrial metabolism of acylcarnitines was affected by sustained Mg^2+^ deficiency in *Trpm6-*deficient adult mice. Since freshly isolated mitochondria are metabolically active for a very short time only (a few hours), we had to acutely deplete Mg^2+^ in the mitochondrial matrix using buffers with low Mg^2+^ concentrations.

When attempting to recapitulate the phenotype of living *Trpm6*-deficient mice at the cellular levels, several critical issues need to be considered: First, according to our model *Trpm6-*mediated Mg^2+^ uptake is not required for cell autonomous functions, implying that *Trpm6*-deficient cells (like TS cells, Figure 6 and Figure 6—figure supplement 1) will not develop Mg^2+^ deficiency and abnormal energy metabolism, because TRPM7 is still functional. Second, *Trpm6* is not expressed in tissues, which critically rely on mitochondrial energy production, such as skeletal muscle or liver. Furthermore, during the isolation process of primary cells it is virtually impossible to “clamp” intracellular Mg^2+^ concentrations at exactly the same levels normally occurring in vivo in *Trpm6*-deficient mice. Accordingly, in vitro experiments with liver or skeletal muscle cells isolated from *Trpm6*-deficient mice would not truly reflect in vivo conditions of *Trpm6*-deficient mice, such as prolonged (several weeks) organismal Mg^2+^ deficiency.

Considering these limitations, we resorted to an alternative experimental model that allowed us to investigate whether Mg^2+^ deficiency in isolated cells affects mitochondrial function. We found that CRISPR/Cas9-mediated inactivation of TRPM7 in the genetically tractable HAP1 cell line (human haploid leukaemia cells) results in Mg^2+^ deficiency and, consequently, a Mg^2+^-dependent proliferation defect (new Figure 6—figure supplement 3). Remarkably, we observed that TRPM7-deficient HAP1 cells display reduced ATP levels at resting conditions. Finally, we show that the respiration rate of TRPM7-deficient HAP1 cells was markedly supressed when compared to control cells (new Figure 6—figure supplement 3). Taken together, we conclude that Mg^2+^ deficiency of HAP1 cells recapitulates our key findings in *Trpm6*-deficient mice. These new findings are now shown in Figure 6—figure supplement 3.

*2) The authors propose that TRPM6 regulates Mg^2+^ transport by reducing the MgATP-dependent inhibition of heteromeric TRPM6/7 channels. First, the evidence for this rests on whether the inhibition curves in Figure 6 are significantly different between WT and KO animals; this should be evaluated. In addition, it is not clear that MgATP in intact cells is in the required range that would differentially affect the current in the WT vs. TRPM6 groups (Figure 6); either the ATP level should be measured, or a suitable reference provided. Finally, another group (Zhang, Yu et al., 2014) reported that TRPM6 expression by itself produced currents, and TRPM6 and M7 coexpression generated currents that were essentially insensitive to MgATP inhibition. This should be discussed in the context of the results shown in Figure 6, where TRPM7 KO completely eliminated current, and wild type and TRPM6 currents are both inhibited by MgATP.*

We agree with the referees and introduced several changes in the manuscript to address their concerns. We re-analyzed the inhibition curves for Mg^2+^ and MgATP using a nonlinear (least-squares) regression fitting and F-test (GraphPad Prism 6.0 software) to address the questions: (i) Is the dose-response dataset of KO cells is statistically different from that in control cells? And (ii) are IC_50_ values distinct for the inhibitory curves obtained for KO and control cells? This analysis supported our initial conclusion that Mg^2+^ elicits similar inhibitory effects on the currents in WT and KO cells, whereas the inhibitory dose-response curves for MgATP were statistically different between KO and control cells. This information is now included in the Results (subsection “TRPM6 cooperates with TRPM7 to regulate divalent cation currents”) and Methods (subsection “Isolation and characterization of mouse trophoblast stem (TS) cells”) sections.

As suggested, we referenced the physiological levels of cytosolic MgATP (subsection “TRPM6 cooperates with TRPM7 to regulate divalent cation currents”) to support the notion that intracellular MgATP can differentially affect ion currents in control and KO cells. In addition, we discuss the work of Zhang et al. (Discussion, fifth paragraph) reporting the functional analysis of recombinant TRPM6 and TRPM7 co-transfected at the ratio 1:1 in HEK 293 cells. A key finding of this study was that recombinant TRPM6 offset the sensitivity of the TRPM7/M6 complex to cytosolic MgATP. Assuming that native currents in WT cells are mediated by TRPM7 homomers and TRPM6/M7 heterotetramers, it is imaginable that genetic ablation of the TRPM6/M7 fraction would only partially reduce the inhibitory effect of MgATP on whole-cell currents. Hence, we conclude that the observations of Zhang et al. are compatible with our results. Another finding from Zhang et al. is that overexpression of TRPM6 homomers in HEK 293 cells results in expression of a functional channel only if TRPM6 cDNA was expressed by the pCINeo-IRES-GFP vector, whereas the same cDNA sequence placed in various other expression plasmids did not produce active TRPM6 channels. This feature of TRPM6 cDNA appears to be unique among TRP channels, and as the nature of this observation is not currently understood, some caution is required in interpreting the results. Therefore, we have concentrated on the functional analysis of endogenous TRPM6 channels in primary cells for defining the cellular role of TRPM6 rather than on overexpression data.

Finally, we would like to emphasize that according to our model, TRPM6 regulates Mg^2+^ uptake by two means: (i) by increasing amplitudes of TRPM7-like currents and (ii) by relieving TRPM7 from the negative feedback by MgATP. To better illustrate the first mechanism, we now include a new Figure 6, showing measurements of native currents in the absence of external divalent cations. This approach is widely used for a quantitative assessment of otherwise very small inward currents of TRPM7/M6, since external divalent cations elicit a strong permeation block of the channel at physiological membrane potentials. As expected, these experiments showed that current amplitudes recorded in KO cells were substantially lower than in control cells.